# ONLINE BOUNDARY-FREE CONTINUAL LEARNING BY SCHEDULED DATA PRIOR

**Hyunseo Koh**[1,2]  **Minhyuk Seo**[1]  **Jihwan Bang**[3,4]  **Hwanjun Song**[3,4]
**Deokki Hong**[1]  **Seulki Park**[5]  **Jung-Woo Ha**[3,4]  **Jonghyun Choi**[1,†]
[1]Yonsei University  [2]GIST  [3]NAVER AI Lab  [4]NAVER Cloud  [5]Seoul National University

## ABSTRACT

Typical continual learning setup assumes that the dataset is split into multiple discrete tasks. We argue that it is less realistic as the streamed data would have no notion of task boundary in real-world data. Here, we take a step forward to investigate more realistic online continual learning – learning continuously changing data distribution without explicit task boundary, which we call *boundary-free* setup. Due to the lack of boundary, it is not obvious when and what information in the past to be preserved for a better remedy for the stability-plasticity dilemma. To this end, we propose a *scheduled* transfer of previously learned knowledge. In addition, we further propose a data-driven balancing between the knowledge in the past and the present in learning objective. Moreover, since it is not straightforward to use the previously proposed forgetting measure without task boundaries, we further propose a novel forgetting and knowledge gain measure based on information theory. We empirically evaluate our method on a Gaussian data stream and its periodic extension, which is frequently observed in real-life data, as well as the conventional disjoint task-split. Our method outperforms prior arts by large margins in various setups, using four benchmark datasets in continual learning literature – CIFAR-10, CIFAR-100, TinyImageNet and ImageNet. Code is available at https://github.com/yonseivnl/sdp.

## 1 INTRODUCTION

In real-world continual learning (CL) scenarios (He et al., 2020), data arrive in a streamed manner (Aljundi et al., 2019a; Cai et al., 2021) whereas typical continual learning setups split the data into multiple discrete *tasks* whose data distributions differ from each other. Moreover, most CL algorithms are studied in an *offline* CL setup (Kirkpatrick et al., 2017; Rebuffi et al., 2017; Saha et al., 2021), where the model can access data multiple times. While being prevalent in the literature, this setup has a number of issues far from the realistic scenario. Although the task setup have been partly addressed by (Prabhu et al., 2020; Koh et al., 2021; Kim et al., 2021b; Bang et al., 2022), the revised setups still have the notion of *task boundary* whereas real-world data may not have the explicit task boundaries as the data distribution changes continuously. Despite that many methods update the model in a boundary-agnostic manner, called *task-free* CL (Aljundi et al., 2019b; Koh et al., 2021), they still leverage the notion of task boundary for knowledge transfer and evaluation, *e.g.*, leveraging the fact that distribution shift in data stream occurs only at task boundaries. In addition, the definition of forgetting depends on the notion of 'old' and 'new' tasks, which are defined by the task boundary.

We argue to address an online CL setup where data are learned online (allowing only a single access to data) with continuous distribution shift without explicit task boundaries. We refer to the setup as *online boundary-free continual learning*. In this setup, a small set of data is streamed to the model one by one, and the model only has access to the current data batch only (Aljundi et al., 2019c;a) without the notion of task boundary.

For the distribution of a continuous data stream, following (Shanahan et al., 2021; Wang et al., 2022), we consider Gaussian distribution as an instance of data streaming distributions. The Gaussian

---

† indicates the corresponding author.

online data stream models the frequency of each class as Gaussian distribution over time. Note that classes do not recur after their initial Gaussian mode in this setup. However, in real-world data, the frequency of a class may have multiple recurring modalities over time, rather than a single mode, as depicted in Fig. 1. To further address such a scenario, we investigate a *periodic*-Gaussian online stream, where each class would recur and the recurrence is periodic. To the best of our knowledge, this is the first work to study CL in continuous data distributions either in periods or not.

The boundary-free setup poses several challenges as follows. In the CL setup with explicit task boundary definition, methods using both episodic memory and distillation (*i.e.*, using data prior (Buzzega et al., 2020; Wu et al., 2019; Hou et al., 2019)) show compelling performance (Masana et al., 2020). They store the model weights at each task boundary and use the stored model as a distillation teacher for mitigating catastrophic forgetting. However, in the continuous data stream, it is challenging to determine which past models should be stored to be used as a data prior. To determine which data prior to transfer the knowledge from, we propose to combine different exponential moving average (EMA) distributions to have a particular schedule of transferring past knowledge.

In addition, as the past knowledge is now from diverse contexts, it is not trivial to balance the supervisory signal from the past and the present. Instead of using a fixed balancing hyperparameter, we propose to learn to balance them for better generalization in multiple scenarios, *i.e.*, datasets. In our empirical studies, we observe that our method outperforms comparable prior arts in Gaussian, periodic-Gaussian, and disjoint task-split data stream on 4 popular benchmarks in CL literature.

Moreover, conventional performance metrics for CL methods including forgetting is not trivially applicable to our setup as they are defined on the task boundary. Here, we propose a new metric for measuring forgetting using information theory. In contrast to the conventional forgetting metric, it captures loss and gain of intra-class knowledge, appropriate for periodic data distribution where the model has to accumulate different knowledge about the same class over multiple periods.

We summarize our contributions as follows:

- Extensively studying online CL with continuous data stream setup without explicit task boundary, including newly proposed periodic CL setup.
- Proposing an online boundary-free CL method that uses scheduled transfer of past knowledge.
- Proposing to learn to balance amount of using past and present knowledge.
- Proposing new metrics that can measure loss of past knowledge (*i.e.*, forgetting) and gain of new knowledge (*i.e.*, the opposite of intransigience) based on information theory.

## 2 RELATED WORK

**Setups for Continual Learning.** With the increasing popularity of CL, there have been several proposals on learning configurations to be realistic. As the first task setup to mimic a real-world data stream that continuously changes over time, prior arts have employed the notion of *task-split*, where the entire data is split into multiple subsets for different continuous tasks (Rebuffi et al., 2017; Castro et al., 2018; Wu et al., 2019). When each subset arrives, it is stored and provided as a set of examples to learn a model. They use multiple epochs to learn a model (Kirkpatrick et al., 2017; Rebuffi et al., 2017; Saha et al., 2021), which is referred as *offline* setup, while a small batch sample of the subset is streamed and used only once in *online* setup (Rolnick et al., 2019; Chaudhry et al., 2019; Aljundi et al., 2019a). In the task-split setup, task boundaries are available and well exploited in various offline/online CL methods (Lopez-Paz & Ranzato, 2017; Rebuffi et al., 2017).

In recent literature, however, there has been efforts to question whether the task-split setup is realistic. To enforce the class distribution of each split differently, *disjoint task-split* confines each class to be assigned to only a single task (Castro et al., 2018). As the disjoint setup is rather artificial as the data stream arrives in class agnostic manner. *Blurry* task-split allows every task shares all classes but with different dominance (Aljundi et al., 2019c; Bang et al., 2021). The *i-Blurry* task-split further guarantee some classes to be added incremental to the blurry task-split (Koh et al., 2021). However, these task configurations have explicit task boundary, which is still artificial. For a more realistic scenario, task-free CL (Aljundi et al., 2019b) has been studied, where models are not allowed to use task boundary information during training. However, they still train and evaluate methods on task-

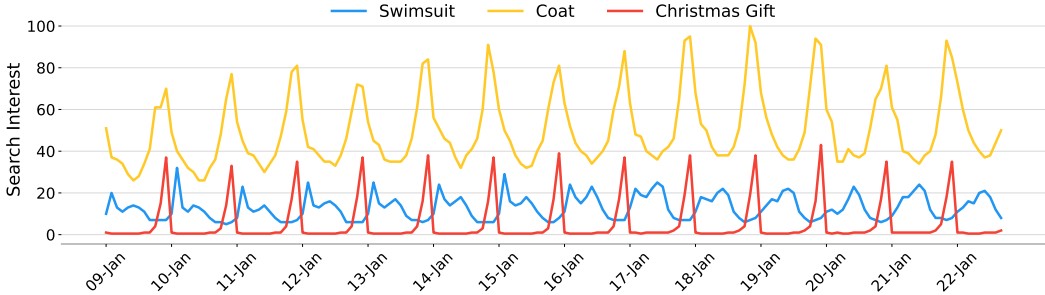

Figure 1: Search interest trends of three items (*i.e.*, swimsuit, coat, Christmas gift) from 2009 to the present in Google Trends. Each item follows a periodic distribution with its own mode and duration.

split setup such as disjoint, blurry, and i-blurry. Thus, task-free is more of a restriction on methods rather than a setup.

In contrast to the prior work, we propose a *boundary-free* setup. It removes the notion of artificial task boundary, rather, the data arrival follows a certain continuous distribution over time. (Shanahan et al., 2021; Wang et al., 2022) investigate data stream following Gaussian distribution, they still split the data into micro-tasks and neglect a periodic property of real-world data, which we study further here.

**Online Continual Learning Methods.**    One of the goals of successful CL learners is not to forget knowledge obtained from the preceding tasks; here, catastrophic forgetting has become the main challenge with deep neural networks for CL (French, 1999). To mitigate the issue, there are four main directions in the recent literature, namely distillation, memory replay, parameter isolation, and regularization. For a more comprehensive review, we refer the reader to surveys (De Lange et al., 2021; Mai et al., 2022).

The key difference between online and offline CL is the accessibility of streaming inputs; the model can only see the entire stream once except for the samples in the episodic memory. To tackle this online constraint, several methods have been developed based on the similar ways in offline CL. GEM (Lopez-Paz & Ranzato, 2017) leverages the gradient of samples in the available memory so that they alleviate forgetting the knowledge of the previous tasks. A-GEM (Chaudhry et al., 2018b) proposes to utilize the average of gradients for each task instead of using the projection of all the gradients. It further saves memory usage and reduces the computational cost. GDumb (Prabhu et al., 2020) proposes greedy balance selection which randomly selects the samples while balancing the number of selected samples per class. GSS (Aljundi et al., 2019c) and RM (Bang et al., 2021) utilize the gradient and uncertainty of each sample, respectively, to increase diversity of selected samples. CLIB (Koh et al., 2021) makes up informative samples in the memory by discarding the least informative samples in the memory for further training.

Unlike offline CL, however, online CL understudied the methods which utilize knowledge distillation. DER (Buzzega et al., 2020) utilized the knowledge distillation between logits of an original image and augmented image from origin, which is well-known and widely used. In our proposed setup, which is *boundary-free*, it is not trivial to determine when and what knowledge should be transferred to mitigate the forgetting. Here, we propose a new knowledge distillation method which is suitable for online boundary-free CL by using scheduled data prior.

## 3    ONLINE TASK BOUNDARY-FREE CONTINUAL LEARNING SETUP

Motivated by a real data stream that changes continuously (*e.g.*, Google search trend of 'swimsuit', 'coat', and 'Christmas gift' during 13 years, depicted in Fig. 1[1]), we propose a new CL setup called *online boundary-free*. We argue that our setup is more realistic than the task-split setup for the following considerations: (1) an ever-changing distribution with (periodic) Gaussian online stream, (2) no notion of explicit task boundaries, and (3) any-time inference for online continual learning.

---

[1] https://trends.google.com/trends/explore?date=all&geo=US&q=christmas%20gift,coat,swimsuit

### 3.1 GAUSSIAN ONLINE STREAM

For easeness of modeling, here we assume that a real data stream for each class follows Gaussian distribution. First, we consider a single mode stream, then extend it to a multi-modal periodic stream. Gaussian-distributed task modeling has been addressed by Shanahan et al. (2021); Wang et al. (2022). However, they differ from ours since their distribution change is not fully continuous as they still split data into multiple micro-tasks and overlook the periodicity of the online stream.

Specifically, we consider the 'arrival time' of samples to be modeled by the Gaussian distribution. In order to create an online stream without task boundary; samples are streamed in an increasing order of their arrival time. Formally, let us assume that the distribution of arrival time for a class $i$ follows the Gaussian distribution $\mathcal{N}(\mu_i, \sigma^2)$. For each class $i$, the mean $\mu_i$ (=mode) of the distribution is exclusively chosen from $\left\{0, \frac{1}{N}, \ldots, \frac{N-1}{N}\right\}$ where $N$ is the number of classes, and the standard deviation $\sigma$ is the same for all classes for simplicity (we use $\sigma = 0.1$ for our empirical validations and provide analysis on different values in the appendix).

**Periodic Gaussian Online Stream.** By a simple modification, we can extend the Gaussian stream to a *periodic* one. Now, the distribution of arrival time of a class $i$ not only follows the Gaussian distribution but also is repeated multiple times:

$$\frac{1}{R} \sum_{r=0}^{R-1} \mathcal{N}(\mu_i + r, \sigma^2), \tag{1}$$

where $R$ is the number of repetition (*i.e.*, periods, we use $R = 5$ in our experiments). The mean $\mu_i$ and standard deviation $\sigma$ are the same as the ones in the non-periodic Gaussian online stream.

## 4 EVALUATION METRICS FOR CONTINUAL LEARNERS IN THE BOUNDARY-FREE DATA STREAM

For evaluating the overall performance in a boundary-free CL, we use the area under the curve of accuracy ($A_{AUC}$) proposed in (Koh et al., 2021) for measuring area under the curve of accuracy-to-(# samples) curve, and last accuracy ($A_{last}$) that measures the final accuracy after learning all samples. We cannot measure $A_{avg}$ metric since there is no task boundary.

Previous CL research use Forgetting and Intransigence metric (Chaudhry et al., 2018a) to investigate stability and plasticity of CL algorithms. However, these metrics require to measure the accuracy of the previous task, so they are not readily measurable in the boundary-free setup due to lack of task boundary. Here, we propose new metrics of measuring forgetting and ability to learn new knowledge by computing loss and gain of knowledge based on information theory.

**Knowledge Loss Ratio.** Specifically, we want to measure loss and gain of knowledge between arbitrary two points in the training, $t_1$ and $t_2$. Let $Y_{GT}$ be ground truth label for a randomly selected sample from a data distribution and $Y_t$ be the model's prediction for that sample at time $t$. We define the Total Knowledge ($TK(t)$) at time $t$ as:

$$TK(t) := I(Y_t; Y_{GT}), \tag{2}$$

where $I(X;Y) = \sum_{y \in \mathcal{Y}} \sum_{x \in \mathcal{X}} P(x,y) \log\left(\frac{P(x,y)}{P(x)P(y)}\right)$ is a mutual information that measures the quantity of information one variable has about the other. In other words, it measures how much information about ground truth we can obtain by observing the model's prediction at time $t$.

To measure the loss of knowledge between $t_1$ and $t_2$, we quantify the knowledge in $Y_{t_1}$ but not in $Y_{t_2}$. The knowledge loss can be measured by the knowledge difference between having both output $(Y_{t_1}, Y_{t_2})$ and having only $Y_{t_2}$. Thus, we define Knowledge Loss ($KL(t_1, t_2)$) between $t_1$ and $t_2$ as:

$$KL(t_1, t_2) := I(Y_{t_1}, Y_{t_2}; Y_{GT}) - I(Y_{t_2}; Y_{GT}) = I(Y_{t_1}; Y_{GT}|Y_{t_2}), \tag{3}$$

where $I(X;Y|Z) = I(X,Z;Y) - I(Z;Y)$ is the conditional mutual information. By dividing the knowledge loss by the total knowledge at $t_1$, we obtain Knowledge Loss Ratio ($KLR(t_1, t_2)$) as:

$$KLR(t_1, t_2) := \frac{KL(t_1, t_2)}{TK(t_1)} = \frac{I(Y_{t_1}; Y_{GT}|Y_{t_2})}{I(Y_{t_1}; Y_{GT})}. \tag{4}$$

KLR measures the ratio of the past knowledge that was lost between $t_1$ and $t_2$, which we will use an equivalent measure for forgetting in continuous data stream.

**Knowledge Gain Ratio.** Similarly, we can define the Knowledge Gain ($KG$) between $t_1$ and $t_2$ as:

$$KG(t_1, t_2) = I(Y_{t_1}, Y_{t_2}; Y_{\text{GT}}) - I(Y_{t_1}; Y_{\text{GT}}) = I(Y_{t_2}; Y_{\text{GT}}|Y_{t_1}). \tag{5}$$

Similar to KLR, we define Knowledge Gain Ratio (KGR) to measure the ratio of the potentially obtainable knowledge obtained between $t_1$ and $t_2$. The amount of information in GT label is $H(Y_{\text{GT}})$ where $H(Y) = -\sum_{y \in \mathcal{Y}} P(y) \log P(y)$ is the entropy, so the potentially obtainable knowledge at time $t_1$ is $H(Y_{\text{GT}}) - TK(t_1)$. Thus, $KGR(t_1, t_2)$ is defined as:

$$KGR(t_1, t_2) := \frac{KG(t_1, t_2)}{H(Y_{\text{GT}}) - TK(t_1)} = \frac{I(Y_{t_2}; Y_{\text{GT}}|Y_{t_1})}{H(Y_{\text{GT}}) - I(Y_{t_1}; Y_{\text{GT}})}. \tag{6}$$

**Implications.** We illustrate the implication of the defined $KL$ and $KG$ in a Venn-Diagram of Fig. 2. Upper circle, lower left circle, lower right circle represent $Y_{\text{GT}}$, $Y_{t_1}$, and $Y_{t_2}$, respectively. KL is knowledge about GT that is in model at $t_1$ but not in $t_2$. KG is the knowledge about GT that is in model at $t_2$ but not in $t_1$. The triple intersection $I(Y_{t_1}; Y_{t_2}; Y_{\text{GT}})$ represents retained knowledge about GT that is in model at both $t_1$ and $t_2$. Finally, the lower intersection, $I(Y_{t_1}; Y_{t_2}|Y_{\text{GT}})$, represents useless information shared in both models that are not relevant to GT labels, such as bias in the model.

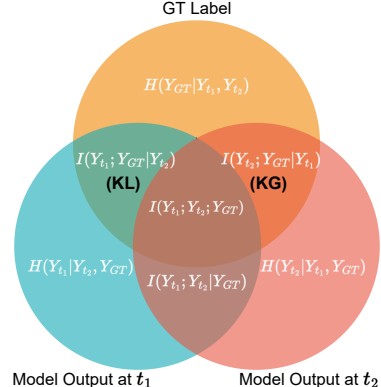

Figure 2: Relation of knowledge learned by model to the Knowledge Loss ($KL$) and Knowledge Gain ($KG$).

One advantage of $KL$ is that it can be interpreted as intraclass forgetting. Since conventional 'forgetting' is measured using task-wise or class-wise accuracy, if forgetting and knowledge gain simultaneously happens within a task or a class, its net effect is zero, thus its effect will not be measured. For example, if we assume there is class $A$ with features $a$ and $b$ and the model forgot about feature $a$ and learned about feature $b$ so that overall accuracy for class $A$ is the same, it counts as zero forgetting in the conventional measure. In contrast, when using the proposed Knowledge Loss ($KL$), the current model only has information about feature $b$, but past and current models combined have information about both $a$ and $b$, so it captures loss of information about feature $a$.

## 5 APPROACH

The existing task-split setup partitions the data stream into multiple discrete tasks. Prior arts in CL literature that have been developed in this setup focus on extracting information from prior models learned in the previous tasks. However, in the online boundary-free setup whose data stream is not partitioned into explicit tasks, it is necessary to consider information at which previous moment we have to transfer to the current time step for better stability and plasiticity trade-off. Although there is no task boundary, a model learned in the past can be stored when each sample arrives and used as a data prior, *i.e.*, a teacher in distillation framework. However, it is not clear which previous models to be used to transfer knowledge to the current time step. We consider various weighting functions to transfer information from the previous data stream in online and continuous fashion, and summarize the empirical results in Table 1.

### 5.1 SCHEDULED DATA PRIOR

To determine the amount of past knowledge to be transferred in a continuous fashion, we consider a schedule of transfer of past knowledge by a composite function of exponential moving average (EMA). EMA calculates a weighted average in an online manner with exponentially decaying weight to place a higher weight on recent datapoints. EMA model $\theta_\alpha(t)$ with EMA ratio $\alpha$ at timestep $t$ is defined recursively as

$$\theta_\alpha(t) = (1 - \alpha) * \theta_\alpha(t - 1) + \alpha * \theta(t) \tag{7}$$

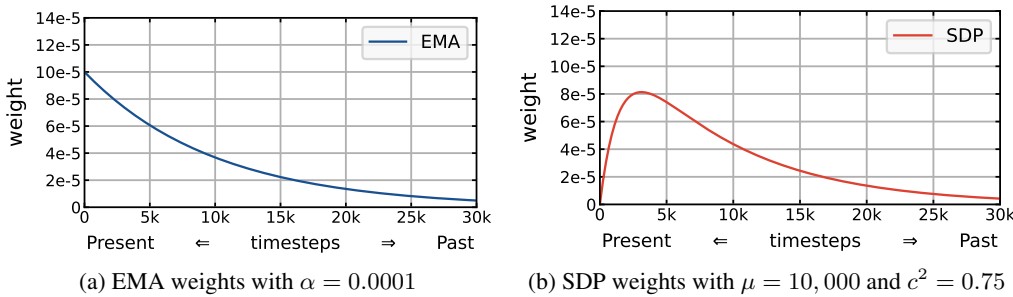

(a) EMA weights with $\alpha = 0.0001$      (b) SDP weights with $\mu = 10,000$ and $c^2 = 0.75$

Figure 3: Comparison between weight distribution of EMA and SDP.

where $\theta(t)$ is the online model's parameter at time $t$. In particular, the EMA update emphasizes the recent knowledge more than the past knowledge by a exponentially decreasing weight distribution, as shown in Fig. 3-(a). However, we argue that recently learned information is well maintained in the model being currently trained (*i.e.*, not yet forgotten) and need to shift the focus to a slightly farther past (as depicted in Fig. 3-(b)).

To implement a such weighting scheme, we propose a hypo-exponential distribution, which has a skewed-bell curve shape to transfer the knowledge from the past. We configure the hypo-exponential distribution by weighted average of two EMA curves with different hyperparameters. The resulting distribution has a mode in the distant past while the vanilla EMA has the mode at the nearest past then the weight monotonically decreases.

Specifically, to construct the scheduled data prior (SDP), we take a weighted sum of two EMA models $\theta_\alpha(t)$ and $\theta_\beta(t)$ with EMA ratios of $\alpha$ and $\beta$ where $\alpha > \beta$. Using the coefficient for the hypo-exponential distribution, the proposed SDP model $\theta_{\text{SDP}(\alpha,\beta)}(t)$ is defined as:

$$\theta_{\text{SDP}(\alpha,\beta)}(t) = \frac{\alpha}{\alpha - \beta}\theta_\beta(t) - \frac{\beta}{\alpha - \beta}\theta_\alpha(t). \tag{8}$$

Its weight are non-negative and summed up to 1, and form a skewed-bell curve as depicted in Fig. 3-(b). Instead of using $\alpha$ and $\beta$, we define SDP with mean $\mu$ and coefficient of variation $c^2 = \frac{\sigma^2}{\mu^2}$ of weight distributions, where $\sigma^2$ is the variance of the distribution, so that hyperparameters are interpretable. The values of $\alpha$ and $\beta$ can be calculated from $\mu$ and $\sigma$ as:

$$\alpha = \frac{1 + \sqrt{2c^2 + \frac{2}{\mu} - 1}}{\mu(1 - c^2) - 1}, \quad \beta = \frac{1 - \sqrt{2c^2 + \frac{2}{\mu} - 1}}{\mu(1 - c^2) - 1}, \tag{9}$$

where $\alpha$ and $\beta$ are positive real values when $\frac{1}{2} - \frac{1}{\mu} < c^2 < 1 - \frac{1}{\mu}$. The derivation for these can be found in appendix.

We empirically validate our hypothesis by a comparative study of using different scheduling distribution of past knowledge transfer in Tab. 1. As shown in the table, the proposed weight distribution (SDP) outperform other scheduling functions by a noticeable margin.

| Teacher Model | $A_{\text{AUC}} \uparrow$ | $A_{last} \uparrow$ |
|---|---|---|
| None (Baseline) | 62.37±0.52 | 64.22±0.97 |
| Periodically Saving Model (Offline) | 63.44±0.10 | 65.67±1.38 |
| Saved Logits (DER) | 59.82±0.30 | 58.92±2.36 |
| Saved Features (DER) | 61.55±0.22 | 58.78±2.38 |
| EMA Model | 63.75±0.22 | 66.55±3.32 |
| SDP Model (Ours) | **64.04±0.25** | **69.47±1.79** |

Table 1: Comparison of past knowledge transfer functions for continual learners on a Non-Periodic Gaussian data stream in CIFAR-10. The proposed method outperforms other functions including EMA Model.

## 5.2 LEARNING TO BALANCE PAST AND PRESENT KNOWLEDGE IN LEARNING

While we transfer the knowledge from the past experience by the SDP distribution, balancing the amount of information from the past and the current time step is a remaining challenge for continual learners (Mai et al., 2022). The balancing hyperparameter is usually tuned by a hand-crafted manner

and is dataset specific. Moreover, fixed parameter is not always optimal, as importance of classification and distillation may vary over the course of training. Thus, we propose to learn to balance them for better generalization to different data context and adapting in different phases of training.

With the learnable balancing parameter $\lambda_t$, we can write the final objective as:

$$\mathcal{L}(x,y) = \lambda_t \cdot \mathcal{L}_{\text{CE}}(x,y) + (1-\lambda_t) \cdot \eta_k \cdot \mathcal{L}_{\text{KD}}(x), \tag{10}$$

where $\mathcal{L}_{\text{CE}}(x,y)$ is the cross-entropy loss for current time step's knowledge and $\mathcal{L}_{\text{KD}}(x) = \| f(x) - f_{\theta_{\text{SDP}}}(x)\|_2^2$ is the L2 distillation loss for the past knowledge transfer and $f_{\theta_{\text{SDP}}}$ is the neural network with parameters $\theta_{\text{SDP}}$ obtained by equation 5. Before balancing the two terms for each task, since model parameter update is proportional to gradient, we normalize the scale of the two terms by $\eta_k$:

$$\eta_k = \frac{|\nabla_f \mathcal{L}_{\text{CE}}|}{|\nabla_f \mathcal{L}_{\text{KD}}|}, \tag{11}$$

where $f$ is the feature layer of the model and $k$ is a batch index. Note that we use the gradient norm at the feature layer since that of the earlier layers will be proportional to it thanks to the chain rule.

$\lambda_t$ is typically defined as $\frac{|C_{\text{new}}|}{|C_{\text{new}}|+|C_{\text{old}}|}$ (Wu et al., 2019), where $C_{\text{new}}$ is the new classes, *i.e.*, classes in the current task and $C_{\text{old}}$ is old classes, *i.e.*, classes from the previous tasks. In the online boundary-free setup, unfortunately, as task is not defined, the notion of new or old is not available for each class. Instead of hard assignment of new and old classes, we measure $0 \leq \gamma_i \leq 1$ that represents how new class $i$ is. The $\gamma_i$ is measured by the inverse of past model's average confidence over samples in class $i$, since the model will have low confidence for new classes. If a past model predicts class $i$ with $p(i) = 1$, we consider that the model has completely learned class $i$, so $\gamma_i = 0$. If it predicts class $i$ with $p(i) \leq \frac{1}{N}$ where $N$ is the total number of classes, *i.e.*, no better than a random model, we consider that the model has learned nothing about class $c$, so $\gamma_i = 1$. To compute $\gamma_i$, we use the samples' confidence on currently learned model at the current time step before using them for training, as a proxy of validation accuracy.

Finally, we define the online balancing parameter $\lambda_t$ by averaging confidence for all the samples for each class in the current time step as:

$$\lambda_t = \frac{\sum_{i=0}^{N-1} \gamma_i}{N}, \quad \text{where} \ \ \gamma_i = \begin{cases} 1, & \text{if } p(i) < 1/N, \\ \frac{N}{N-1}\left(1-p(i)\right), & \text{otherwise.} \end{cases} \tag{12}$$

## 6 EXPERIMENTS

**Experimental Setup.** For empirical validations, following (Koh et al., 2021; Guo et al., 2022), we use four benchmark datasets; CIFAR-10, CIFAR-100, TinyImageNet, and ImageNet. Most of our experiments are on Gaussian and Periodic-Gaussian online stream defined in Sec. 3.1 for boundary-free setup except in Sec. 6.3 where we evaluate methods on disjoint stream using 5-split-CIFAR. We use $\sigma = 0.1$ for Gaussian setup, and $\sigma = 0.1$, $R = 5$ for Periodic-Gaussian setup.

For evaluation metrics, we use the area under the curve of accuracy ($A_{AUC}$) and last accuracy ($A_{last}$) for overall performance, and knowledge loss ratio (KLR) and knowledge gain ratio (KGR) for stability and plasticity, as defined in section 4. KLR and KGR are measured every $10,000$ samples for CIFAR-10 and CIFAR-100, every $20,000$ samples for TinyImageNet, every $100,000$ samples for ImageNet. All results are averaged over 3 different random seeds, except ImageNet (Bang et al., 2021; Koh et al., 2021) for computational cost. We will publicly release the implementation of our method and the continually learned models.

**Method Detail.** SDP uses an episodic memory, updated by the Greedy Balancing Sampler (Prabhu et al., 2020). For training, we use only the samples randomly selected from the memory, following (Koh et al., 2021). We provide a pseudocode of SDP in Appendix Sec. A.4. For the hyperparameters, we use $\mu = 10,000$ and $c^2 = 0.75$ found by hyperparameter search on CIFAR-10 non-periodic Gaussian setup, for all experiments. Details can be found in Appendix Sec. A.9

**Baselines.** We compare our method with other baselines that can be used in task-free setup: ER (Rolnick et al., 2019), DER++ (Buzzega et al., 2020), ER-MIR (Aljundi et al., 2019a), Gdumb (Prabhu et al., 2020), and CLIB (Koh et al., 2021). Note that since our setup is boundary-free, only the task-free methods can be applied.

| Methods | CIFAR-10 | | | | CIFAR-100 | | | |
|---|---|---|---|---|---|---|---|---|
| | $A_{AUC}\uparrow$ | $A_{last}\uparrow$ | $KLR_{avg}\downarrow$ | $KGR_{avg}\uparrow$ | $A_{AUC}\uparrow$ | $A_{last}\uparrow$ | $KLR_{avg}\downarrow$ | $KGR_{avg}\uparrow$ |
| ER | 55.97±0.85 | 58.59±2.56 | 23.36±1.39 | 21.52±0.92 | 36.93±0.77 | 40.93±0.35 | 51.77±0.87 | 17.42±0.50 |
| DER++ | 56.12±0.14 | 60.54±2.37 | 23.60±2.12 | 22.57±0.98 | 28.77±0.67 | 34.48±0.62 | 48.83±1.49 | 17.35±0.16 |
| ER-MIR | 56.48±0.18 | 59.38±2.83 | 25.04±1.67 | 21.72±0.16 | 34.76±1.20 | 37.20±2.06 | 57.23±1.14 | 16.46±0.68 |
| GDumb | 46.45±0.68 | 42.62±0.64 | 35.47±0.51 | 11.69±0.32 | 28.81±0.74 | 25.38±0.26 | 86.75±2.07 | 8.54±0.34 |
| CLIB | 63.01±0.31 | 65.24±2.10 | 20.79±1.64 | 22.52±1.92 | 42.79±0.65 | 43.81±0.58 | 48.59±1.78 | 15.96±0.26 |
| SDP (Ours) | **66.51±0.42** | **76.29±0.50** | **10.75±1.48** | **23.19±1.16** | **46.34±0.64** | **54.93±0.56** | **36.56±0.92** | **18.77±0.27** |

| Methods | TinyImageNet | | | | ImageNet | | | |
|---|---|---|---|---|---|---|---|---|
| | $A_{AUC}\uparrow$ | $A_{last}\uparrow$ | $KLR_{avg}\downarrow$ | $KGR_{avg}\uparrow$ | $A_{AUC}\uparrow$ | $A_{last}\uparrow$ | $KLR_{avg}\downarrow$ | $KGR_{avg}\uparrow$ |
| ER | 21.78±0.76 | 22.64±0.44 | 66.39±1.36 | 9.04±0.88 | 24.16 | 26.64 | 39.43 | 6.19 |
| DER++ | 17.13±0.88 | 16.85±0.63 | 69.65±4.21 | 7.51±0.64 | 19.34 | 21.17 | 40.64 | 5.71 |
| ER-MIR | 21.64±0.77 | 22.55±0.66 | 66.51±1.18 | **9.43±0.46** | 12.86 | 24.70 | 54.53 | 8.23 |
| GDumb | 18.81±0.64 | 15.08±0.25 | 91.58±2.15 | 3.60±0.10 | 13.74 | 10.24 | 64.23 | 2.95 |
| CLIB | 27.53±0.94 | 26.10±0.93 | 63.97±2.18 | 6.16±0.39 | 34.11 | 31.60 | **36.45** | 6.20 |
| SDP (Ours) | **30.49±0.69** | **32.59±0.20** | **54.85±0.30** | 8.30±0.10 | **35.52** | **33.94** | 39.37 | **8.25** |

Table 2: Accuracy of continually learned model in **Non-Periodic Gaussian** data stream in CIFAR-10, CIFAR-100, TinyImageNet and ImageNet. The results except ImageNet are the average values of the three random seeds.

**Implementation Detail.** We use ResNet-18 (He et al., 2016) as the network architecture for all experiments. We set training hyperparameters following (Koh et al., 2021; Bang et al., 2021; Prabhu et al., 2020). For CIFAR-10, CIFAR-100, TinyImageNet and ImageNet, we use batch size of 16, 16, 32, 256, number of updates per sample of 1, 3, 3, 0.25, memory size of 500, 2000, 4000, 20000, respectively. We use Adam optimizer with LR of 0.0003 for all datasets and setup. Constant LR schedule is applied unless the method specifies the LR schedule (Koh et al., 2021; Prabhu et al., 2020). For data augmentation, some prior works (Koh et al., 2021; Bang et al., 2021) use AutoAugment (Cubuk et al., 2019a). However, AutoAugment requires searching the augmentation policy using the dataset, so it is not applicable for our setup. Instead, we use CutMix (Yun et al., 2019) and RandAugment (Cubuk et al., 2019b) with fixed hyperparameters for requiring no policy search.

## 6.1 RESULTS ON NON-PERIODIC GAUSSIAN DATA STREAM

As shown in Table 2, our method outperforms the other baselines on all benchmark datasets. Especially, we can observe that the last accuracy ($A_{last}$) is better than the other baselines by large margins, which verifies that our method can quickly adapt to the current task in online boundary-free setup. In addition, the lower $KLR$ score shows that our method is more robust to forgetting.

## 6.2 RESULTS ON PERIODIC GAUSSIAN DATA STREAM

The results of our method and other baselines on periodic-gaussian online stream is displayed in Table 3. It is observed that our method is superior to other baselines on various datasets with higher accuracies ($A_{AUC}$, $A_{last}$). It is noteworthy that our forgetting scores ($KLR$) is considerably lower compared to other baselines. We believe that our method can transfer information from the appropriate past, and thus be adventageous when the data distribution repeats periodically.

## 6.3 RESULTS ON DISJOINT TASK SPLIT

As our method is not specifically designed for the boundary-free task setup, we also compare our method to prior arts in the disjoint setup. We summarize the results on 5-split-CIFAR-10 (*i.e.*, partitions 10 classes into 5 tasks) and 5-split-CIFAR-100 in Table 4 on the conventional task-split setup. As shown in the table, the proposed SDP outperforms other methods even in the setups with clear task boundaries.

## 6.4 ABLATION STUDY

We now ablate the model for the two components, and summarize the results in Table 5. We observe that both the scheduling function and adaptive loss balancing contribute in improving the performance. Baseline is a simple episodic-memory based method using Greedy Balancing Sampler (Prabhu et al., 2020) and memory-only training (Koh et al., 2021). By using SDP distillation,

| Methods | CIFAR-10 | | | | CIFAR-100 | | | |
|---|---|---|---|---|---|---|---|---|
| | $A_{\text{AUC}}\uparrow$ | $A_{\text{last}}\uparrow$ | $KLR_{\text{avg}}\downarrow$ | $KGR_{\text{avg}}\uparrow$ | $A_{\text{AUC}}$ | $A_{\text{last}}\uparrow$ | $KLR_{\text{avg}}\downarrow$ | $KGR_{\text{avg}}\uparrow$ |
| ER | 61.17±0.97 | 67.29±2.58 | 17.87±0.48 | 22.44±1.05 | 37.90±0.31 | 46.45±0.42 | 48.83±1.49 | 17.35±0.16 |
| DER++ | 60.70±1.07 | 70.41±1.83 | 16.47±0.53 | 23.45±0.49 | 23.07±1.22 | 37.88±1.42 | 48.72±1.49 | 15.56±0.50 |
| ER-MIR | 60.16±1.70 | 67.86±1.06 | 18.49±1.01 | 22.37±0.91 | 35.87±0.28 | 45.08±0.35 | 51.26±0.60 | 16.73±0.12 |
| GDumb | 43.01±0.45 | 42.97±0.91 | 39.88±1.90 | 12.87±0.83 | 25.01±0.13 | 25.37±1.91 | 88.84±1.91 | 7.82±0.11 |
| CLIB | 66.18±0.64 | 73.86±1.38 | 16.52±0.43 | **24.99±0.69** | 42.48±0.54 | 50.91±1.01 | 46.94±0.82 | 17.08±0.27 |
| SDP (Ours) | **69.68±0.33** | **78.66±0.91** | **5.54±0.22** | 18.54±1.28 | **46.90±0.32** | **55.62±0.16** | **33.30±0.41** | **21.41±0.08** |

| Methods | TinyImageNet | | | | ImageNet | | | |
|---|---|---|---|---|---|---|---|---|
| | $A_{\text{AUC}}\uparrow$ | $A_{\text{last}}\uparrow$ | $KLR_{\text{avg}}\downarrow$ | $KGR_{\text{avg}}\uparrow$ | $A_{\text{AUC}}\uparrow$ | $A_{\text{last}}\uparrow$ | $KLR_{\text{avg}}\downarrow$ | $KGR_{\text{avg}}\uparrow$ |
| ER | 20.67±0.25 | 24.84±0.49 | 66.65±1.17 | 7.14±0.12 | 25.25 | 34.93 | 43.04 | 9.39 |
| DER++ | 16.67±0.34 | 21.67±0.86 | 66.39±1.36 | 7.63±0.13 | 20.57 | 27.85 | 44.96 | 9.87 |
| ER-MIR | 20.74±0.36 | 24.57±1.00 | 68.40±1.72 | 7.10±0.06 | 19.52 | 31.85 | 44.73 | 8.65 |
| GDumb | 15.75±0.12 | 15.40±0.76 | 95.73±3.75 | 3.15±0.17 | 11.22 | 11.38 | 67.32 | 2.59 |
| CLIB | 25.45±0.39 | 30.77±1.40 | 61.24±1.37 | 7.47±0.16 | 32.90 | 38.96 | 39.37 | 8.25 |
| SDP (Ours) | **30.14±0.32** | **37.55±0.43** | 49.95±0.30 | **10.68±0.09** | **34.21** | **39.35** | **33.38** | **10.34** |

Table 3: Accuracy of continually learned model in **Periodic Gaussian** data stream. The results except ImageNet are the average values of the three random seeds.

| Methods | CIFAR-10 | | | | CIFAR-100 | | | |
|---|---|---|---|---|---|---|---|---|
| | $A_{\text{AUC}}\uparrow$ | $A_{\text{last}}\uparrow$ | $KLR_{\text{avg}}\downarrow$ | $KGR_{\text{avg}}\uparrow$ | $A_{\text{AUC}}$ | $A_{\text{last}}\uparrow$ | $KLR_{\text{avg}}\downarrow$ | $KGR_{\text{avg}}\uparrow$ |
| ER | 73.52±0.87 | 58.95±2.11 | 24.95±1.41 | 25.37±1.21 | 49.21±1.05 | 39.86±0.52 | 58.02±0.97 | 13.83±0.45 |
| DER++ | 71.01±1.26 | 55.37±2.47 | 30.02±3.22 | 23.62±1.68 | 46.03±1.21 | 37.60±0.75 | 59.95±1.18 | 13.79±0.63 |
| ER-MIR | 71.69±0.91 | 56.01±2.06 | 28.68±4.43 | 23.69±0.52 | 48.48±1.19 | 37.60±0.34 | 62.95±1.31 | 12.88±0.32 |
| GDumb | 62.23±0.70 | 44.69±1.11 | 25.64±1.45 | 11.44±0.83 | 37.80±0.31 | 24.77±0.68 | 79.33±3.13 | 9.04±0.14 |
| CLIB | 73.62±0.87 | 56.21±3.30 | 28.68±4.43 | 23.69±0.52 | 50.26±1.06 | 40.05±0.71 | 59.48±2.00 | 12.23±0.46 |
| SDP (Ours) | **76.89±1.22** | **66.19±3.48** | **15.86±1.74** | **25.68±2.61** | **55.94±1.55** | **50.26±1.41** | **46.35±2.44** | **16.04±0.80** |

Table 4: Results of disjoint task split on CIFAR-10 / 100. Each of datasets are split into 5 tasks

we have 1.7∼5.6% improved $A_{AUC}$ and 5.2∼8.4% improved $A_{last}$ compared to baseline. Since we didn't apply adaptive loss balancing, we use a fixed balancing parameter optimized by grid search on CIFAR-10 non-periodic Gaussian stream. By applying adaptive loss balancing, we have further improved performance, except for $A_{last}$ in CIFAR-100 periodic-Gaussian setup. It still outperforms baseline and other methods by large margins.

| Methods | CIFAR-10 | | | | CIFAR-100 | | | |
|---|---|---|---|---|---|---|---|---|
| | Gaussian | | Periodic-Gaussian | | Gaussian | | Periodic-Gaussian | |
| | $A_{\text{AUC}}\uparrow$ | $A_{last}\uparrow$ | $A_{\text{AUC}}\uparrow$ | $A_{last}\uparrow$ | $A_{\text{AUC}}\uparrow$ | $A_{last}\uparrow$ | $A_{\text{AUC}}\uparrow$ | $A_{last}\uparrow$ |
| Baseline | 62.37±0.52 | 64.22±0.97 | 64.95±0.60 | 72.55±0.71 | 42.02±0.58 | 43.19±0.32 | 40.54±0.23 | 48.15±0.39 |
| + Scheduling Function | 64.04±0.21 | 69.47±1.79 | 67.79±0.21 | 77.87±0.24 | 45.73±0.40 | 50.53±0.18 | 46.14±0.05 | **56.56±0.36** |
| + Data-driven Balancing | **66.51±0.42** | **76.29±0.50** | **69.68±0.33** | **78.66±0.91** | **46.34±0.64** | **54.93±0.56** | **46.90±0.32** | 55.62±0.16 |

Table 5: Ablation Study

# 7 CONCLUSION

For working in a better realistic continual learning scenario, we propose a continual learning setup of a continuous data stream without explicity task boundary defined, called *boundary-free* continual learning, with the constraint of learning the data in online manner. We also investigate the periodicity of the data distribution. As the existing evaluation metrics for continual learning setup assumes the explicit task boundaries, they are not readily applicable in the boundary-free setup. Thus, we propose two new information theoretic evaluation metrics to quantify the amount of information loss and gain over the data stream, named knowledge loss and knowledge gain. To address the continuous data stream to incrementally update the model, we propose a method to leverage the previously learned knowledge by a skewed bell shaped weighting function in a distillation framework, named *scheduled data prior*. For further generalization to different benchmarks, we propose to balance the pace of learning of knowledge in the past and the present in a data-driven manner. In our empirical evaluations, the proposed method outperforms comparable prior arts that update the weights per each data batch in a stream in the proposed boundary-free setup as well as the conventional disjoint task-split setup.

**Limitations.** Since our method does not leverage any notion of periodicity, it is expected to improve the method by considering the fact that the data stream has a periodic, though we may not know the duration of the period.

## ETHICS STATEMENT

Continual learning (CL) including our approach aims to address real-world scenarios where data distributions continuously change in a non-stationary manner. Therefore, it can alleviate some ethical limitation of models resulting from lack of latest knowledge by continuous training and evaluation on newly observed data. This effort can be more effective to handle recent large language models (Brown et al., 2020; Kim et al., 2021a; Chowdhery et al., 2022) or foundational vision models (Bommasani et al., 2021). On the other hand, CL methods might be vulnerable to model bias (Rae et al., 2021) caused by various bias inherent in the changing data due to its continuous tracking. This ethical issue is a future research topic. This may expose the public to discrimination by the deployed deep models due to unsolved issues in deep learning such.

## REPRODUCIBILITY STATEMENT

We take the reproducibility in deep learning very seriously and highlight some of the contents in the manuscript that might help in reproducing our work. We will definitely release our implementations, learned models and newly derived datasets used in our experiments as mentioned in Sec. 6. Also, we have included relevant implementation details in Sec. 6. Finally, we present our final optimization objective (Eq. 10) with necessary details to reproduce the methodology.

## ACKNOWLEDGEMENT

This work is partly supported by the NRF grant (No.2022R1A2C4002300), IITP grants (No.2020-0-01361-003, AI Graduate School Program (Yonsei University) 5%, No.2021-0-02068, AI Innovation Hub 5%, 2022-0-00077, 20%, 2022-0-00113, 20%, 2022-0-00959, 15%, 2022-0-00871, 15%, 2022-0-00951, 15%) funded by the Korea government (MSIT).

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

# A APPENDIX

## A.1 DETAILS ABOUT GAUSSIAN AND PERIODIC GAUSSIAN DATA STREAM SETUP

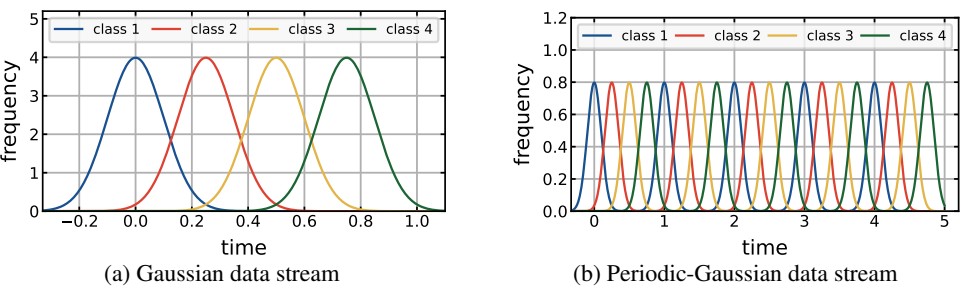

(a) Gaussian data stream          (b) Periodic-Gaussian data stream

Figure 4: Distribution for (a) Gaussian and (b) Periodic Gaussian data streams

The resulting distribution with $N = 4$ is depicted in Fig. 4-(a). The periodic Gaussian distributions with $N = 4$ and $R = 5$ are illustrated in Fig. 4-(b).

## A.2 DERIVATION OF SDP RATIOS FROM MEAN AND VARIANCE

The weights of EMA is same as the pmf of geometric distribution, with $k$-th weight $w_\alpha(k)$ obtained as:

$$w_\alpha(k) = \alpha(1 - \alpha)^{k-1} \tag{13}$$

when $\alpha$ is the EMA ratio. Using the known mean and variance of geometric distribution, Thus, mean $\mu_\alpha$ of EMA is obtained as:

$$\mu_\alpha = \sum_k^\infty k w_\alpha(k) = \frac{1}{\alpha} \tag{14}$$

The variance $\sigma_\alpha^2$ is:

$$\sigma_\alpha^2 = \sum_k^\infty k^2 w_\alpha(k) - \mu_\alpha^2 = \frac{1 - \alpha}{\alpha^2} \tag{15}$$

Thus,

$$\sum_k^\infty k^2 w_\alpha(k) = \frac{1 - \alpha}{\alpha^2} + \frac{1}{\alpha^2} = \frac{2 - \alpha}{\alpha^2} \tag{16}$$

Using the definition of SDP . 5, we obtain $w_{(\alpha,\beta)}(k)$, the $k$-th weights of SDP with two ratios $\alpha$ and $\beta$ as:

$$w_{(\alpha,\beta)}(k) = \frac{\alpha}{\alpha - \beta} w_\beta(k) - \frac{\beta}{\alpha - \beta} w_\alpha(k) \tag{17}$$

The mean $\mu_{(\alpha,\beta)}$ of SDP is calculated as:

$$
\begin{aligned}
\mu_{(\alpha,\beta)} &= \sum_k^\infty k w_{(\alpha,\beta)}(k) \\
&= \frac{\alpha}{\alpha - \beta} \sum_k^\infty k w_\beta(k) - \frac{\beta}{\alpha - \beta} \sum_k^\infty k w_\alpha(k) \\
&= \frac{\alpha}{\beta(\alpha - \beta)} - \frac{\beta}{\alpha(\alpha - \beta)} \\
&= \frac{\alpha + \beta}{\alpha\beta} = \frac{1}{\alpha} + \frac{1}{\beta}
\end{aligned}
\tag{18}
$$

Also, the variance $\sigma^2_{(\alpha,\beta)}$ of SDP is calculated as:

$$\sigma^2_{(\alpha,\beta)} = \sum_k^\infty k^2 w_{(\alpha,\beta)}(k) - \mu^2_{(\alpha,\beta)}$$

$$= \frac{\alpha}{\alpha-\beta} \sum_k^\infty k^2 w_\beta(k) - \frac{\beta}{\alpha-\beta} \sum_k^\infty k^2 w_\alpha(k) - \mu^2_{(\alpha,\beta)}$$

$$= \frac{\alpha(2-\beta)}{\beta^2(\alpha-\beta)} - \frac{\beta(2-\alpha)}{\alpha^2(\alpha-\beta)} - \frac{(\alpha+\beta)^2}{\alpha^2\beta^2}$$

$$= \frac{\alpha^2 + \beta^2 - \alpha\beta(\alpha+\beta)}{\alpha^2\beta^2} \tag{19}$$

Thus, the squared coefficient of variation $c^2 = \frac{\sigma^2}{\mu^2}$ is:

$$c^2 = \frac{(\alpha+\beta)^2 - \alpha\beta(\alpha+\beta+2)}{(\alpha+\beta)^2} \tag{20}$$

Recall

$$\mu_{(\alpha,\beta)} = \frac{\alpha+\beta}{\alpha\beta} \tag{21}$$

Solving these equations for $\alpha + \beta$ and $\alpha\beta$ by substitution, we get

$$\alpha\beta = \frac{2}{c^2 - \mu^2 + 1} \tag{22}$$

$$\alpha + \beta = \frac{2\mu}{c^2 - \mu^2 + 1} \tag{23}$$

Solving this equations for $\alpha$ and $\beta$ using quadratic formula, we get:

$$\alpha = \frac{1 + \sqrt{2c^2 + \frac{2}{\mu} - 1}}{\mu(1-c^2) - 1}, \quad \beta = \frac{1 - \sqrt{2c^2 + \frac{2}{\mu} - 1}}{\mu(1-c^2) - 1}. \tag{24}$$

### A.3 ANALYSIS ON STANDARD DEVIATION OF GAUSSIAN DISTRIBUTION

| Methods | Sigma=0 | | | | Sigma=10 | | | |
|---|---|---|---|---|---|---|---|---|
| | $A_{\text{AUC}} \uparrow$ | $A_{\text{last}} \uparrow$ | $KLR_{\text{avg}} \downarrow$ | $KGR_{\text{avg}} \uparrow$ | $A_{\text{AUC}} \uparrow$ | $A_{\text{last}} \uparrow$ | $KRL_{\text{avg}} \downarrow$ | $KGR_{\text{avg}} \uparrow$ |
| ER | 73.52±0.87 | 58.95±2.11 | 24.95±1.41 | 25.37±1.21 | 55.97±0.85 | 58.59±2.56 | 23.36±1.39 | 21.52±0.92 |
| DER++ | 71.01±1.26 | 55.37±2.47 | 30.02±3.22 | 23.62±1.68 | 56.12±0.14 | 60.54±2.37 | 23.60±2.12 | 22.57±0.98 |
| ER-MIR | 71.69±0.91 | 56.01±2.06 | 28.68±4.43 | 23.69±0.52 | 56.48±0.18 | 59.38±2.83 | 25.04±1.67 | 21.72±0.16 |
| GDumb | 62.23±0.70 | 44.69±1.11 | 25.64±1.45 | 11.44±0.83 | 46.45±0.68 | 42.62±0.64 | 35.47±0.51 | 11.69±0.32 |
| CLIB | 73.62±0.87 | 56.21±3.30 | 28.68±4.43 | 23.69±0.52 | 63.01±0.31 | 65.24±2.10 | 20.79±1.64 | 22.52±1.92 |
| SDP (Ours) | **76.89±1.22** | **66.19±3.48** | **15.86±1.74** | **25.68±2.61** | **66.51±0.42** | **76.29±0.50** | **10.75±1.48** | **23.19±1.16** |

| Methods | Sigma=25 | | | | Sigma=50 | | | |
|---|---|---|---|---|---|---|---|---|
| | $A_{\text{AUC}} \uparrow$ | $A_{\text{last}} \uparrow$ | $KLR_{\text{avg}} \downarrow$ | $KGR_{\text{avg}} \uparrow$ | $A_{\text{AUC}} \uparrow$ | $A_{\text{last}} \uparrow$ | $KLR_{\text{avg}} \downarrow$ | $KGR_{\text{avg}} \uparrow$ |
| ER | 60.42±0.80 | 68.74±2.45 | 18.32±1.42 | 26.89±1.81 | 58.25±0.85 | 58.59±2.56 | 23.36±1.39 | 21.52±0.92 |
| DER++ | 58.25±0.73 | 68.58±0.79 | 16.96±1.67 | **27.49±2.10** | 64.39±0.91 | 75.82±1.38 | 12.56±0.88 | **26.39±0.49** |
| ER-MIR | 60.94±0.92 | 69.21±0.49 | 17.94±0.99 | 25.67±0.93 | 66.82±0.44 | 76.38±0.31 | 13.84±0.11 | 25.20±0.69 |
| GDumb | 41.19±0.18 | 41.79±0.82 | 43.04±2.11 | 12.45±0.24 | 41.65±0.16 | 42.62±0.59 | 39.20±2.20 | 12.98±0.75 |
| CLIB | 66.39±0.09 | 71.41±1.32 | 19.74±0.97 | 26.94±0.69 | 69.62±0.42 | 72.31±1.21 | 7.42±1.12 | 12.87±1.51 |
| SDP (Ours) | **69.97±0.37** | **81.95±0.89** | **4.23±0.64** | 20.67±2.30 | **73.44±0.26** | **82.33±0.26** | **3.57±0.26** | 16.24±0.59 |

Table 6: Comparison of Accuracy for different values of standard deviation for Gaussian data Stream in CIFAR-10

We test CL methods on 4 different values of standard deviation for Gaussian data stream in CIFAR-10 and CIFAR-100. SDP outperform other methods in all tested values of $sigma$, showing that our model is robust to how spread out the distribution is.

| Methods | Sigma=0 | | | | Sigma=10 | | | |
|---|---|---|---|---|---|---|---|---|
| | $A_{\text{AUC}} \uparrow$ | $A_{\text{last}} \uparrow$ | $KLR_{\text{avg}} \downarrow$ | $KGR_{\text{avg}} \uparrow$ | $A_{\text{AUC}} \uparrow$ | $A_{\text{last}} \uparrow$ | $KLR_{\text{avg}} \downarrow$ | $KGR_{\text{avg}} \uparrow$ |
| ER | 49.21±1.05 | 39.86±0.52 | 58.02±0.97 | 13.83±0.45 | 36.93±0.77 | 40.93±0.35 | 51.77±0.87 | 17.42±0.50 |
| DER++ | 46.03±1.29 | 37.60±0.75 | 59.95±1.18 | 13.79±0.63 | 28.77±0.67 | 34.48±0.62 | 48.83±1.49 | 17.35±0.16 |
| ER-MIR | 48.48±1.19 | 37.60±0.34 | 62.95±1.31 | 12.88±0.32 | 34.76±1.20 | 37.20±2.06 | 57.23±1.14 | 16.46±0.68 |
| GDumb | 37.80±0.31 | 24.77±0.68 | 79.33±3.13 | 9.04±0.14 | 28.81±0.74 | 25.38±0.26 | 86.75±2.07 | 8.54±0.34 |
| CLIB | 50.26±1.06 | 40.05±0.71 | 59.48±2.00 | 12.23±0.46 | 42.79±0.65 | 43.81±0.58 | 48.59±1.78 | 15.96±0.26 |
| SDP (Ours) | **55.94±1.55** | **50.26±1.41** | **46.35±2.44** | **16.04±0.80** | **46.34±0.64** | **54.93±0.56** | **36.56±0.92** | **18.77±0.27** |

| Methods | Sigma=25 | | | | Sigma=50 | | | |
|---|---|---|---|---|---|---|---|---|
| | $A_{\text{AUC}} \uparrow$ | $A_{\text{last}} \uparrow$ | $KLR_{\text{avg}} \downarrow$ | $KGR_{\text{avg}} \uparrow$ | $A_{\text{AUC}} \uparrow$ | $A_{\text{last}} \uparrow$ | $KLR_{\text{avg}} \downarrow$ | $KGR_{\text{avg}} \uparrow$ |
| ER | 49.21±1.05 | 39.86±0.52 | 58.02±0.97 | 13.83±0.45 | 32.86±0.39 | 45.11±1.11 | 50.75±1.05 | 17.38±0.43 |
| DER++ | 46.03±1.29 | 37.60±0.75 | 59.95±1.18 | 13.79±0.63 | 31.25±1.59 | 46.00±1.65 | 47.76±0.64 | 17.57±0.06 |
| ER-MIR | 48.48±1.19 | 37.60±0.34 | 62.95±1.31 | 12.88±0.32 | 31.85±0.17 | 44.60±0.17 | 53.32±0.98 | 16.40±0.30 |
| GDumb | 37.80±0.31 | 24.77±0.68 | 79.33±3.13 | 9.04±0.14 | 23.98±0.13 | 24.30±0.54 | 90.32±1.50 | 7.66±0.17 |
| CLIB | 40.06±0.20 | 46.90±0.87 | 49.71±0.60 | 16.51±0.09 | 50.26±1.06 | 40.05±0.71 | 59.48±2.00 | 12.23±0.46 |
| SDP (Ours) | **43.88±0.57** | **55.32±0.50** | **32.92±0.68** | **20.00±0.24** | **46.50±0.42** | **55.97±0.33** | **32.89±0.25** | **21.57±0.10** |

Table 7: Comparison of Accuracy for different values of standard deviation for Gaussian data Stream in CIFAR-100

---

**Algorithm 1** Pseudocode for SDP

1: **Input** model $f_\theta$, EMA models $f_{\theta_\alpha}$, $f_{\theta_\beta}$, EMA ratios $\alpha$, $\beta$, Memory $\mathcal{M}$, Training data stream $\mathcal{D}$, Learning rate $\mu$
2: $\theta_\alpha \leftarrow \theta$, $\theta_\beta \leftarrow \theta$      ▷ EMA models initialize
3: **for** $(x, y) \in \mathcal{D}$ **do**      ▷ sample arrives from training data stream D
4:    **Update** $M \leftarrow$ GreedyBalancingSampler $(\mathcal{M}, (x, y))$      ▷ Update Memory
5:    $\theta_{\text{SDP}}(t) = \frac{\alpha}{\alpha - \beta} \theta_\beta(t) - \frac{\beta}{\alpha - \beta} \theta_\alpha(t).$      ▷ Calculate SDP model parameters
6:    **Update** $\gamma_y \leftarrow$ UpdateConfidenceMean $(f_{\theta_{\text{SDP}}}(x), y)$      ▷ Update class-wise confidence
7:    **Update** $\lambda_t \leftarrow \frac{\sum_{i=0}^{N-1} \gamma_i}{N}$      ▷ Update balancing parameter $\lambda_t$
8:    **Sample** $(X, Y) \leftarrow$ RandomSample$(\mathcal{M})$      ▷ Get batch $(X, Y)$ from Memory
9:    $\mathcal{L}_{\text{CE}}(X, Y) = \text{CrossEntropyLoss}(f_\theta X, Y)$      ▷ Calculate cross-entropy loss
10:    $\mathcal{L}_{\text{KD}}(X) = \|f(X) - f_{\theta_{\text{SDP}}}(X)\|_2^2$      ▷ Calculate distillation loss
11:    $\eta_k = \frac{|\nabla_f \mathcal{L}_{\text{CE}}|}{|\nabla_f \mathcal{L}_{\text{KD}}|}$      ▷ Obtain batch balancing factor $\eta_k$
12:    $\mathcal{L}(X, Y) = \lambda_t \cdot \mathcal{L}_{\text{CE}}(X, Y) + (1 - \lambda_t) \cdot \eta_k \cdot \mathcal{L}_{\text{KD}}(X)$      ▷ Calculate total loss
13:    $\theta \leftarrow \theta - \mu \cdot \nabla_\theta \mathcal{L}(X, Y)$      ▷ Update model
14:    **Update** $\theta_\alpha \leftarrow (1 - \alpha) \cdot \theta_\alpha + \alpha \cdot \theta$      ▷ Update EMA models
15:    **Update** $\theta_\beta \leftarrow (1 - \beta) \cdot \theta_\beta + \beta \cdot \theta$
16: **end for**
17: **Output** $f_\theta$

---

### A.4 PSEUDOCODE FOR THE SDP FRAMEWORK

Algorithm 1 provides detailed pseudocode for SDP. When the sample enters in time order, the memory is updated by Greedy Balancing Sampler (Prabhu et al., 2020). In addition, update the balancing parameter $\lambda$ based on the confidence obtained by running inference on new samples using SDP model. SDP model parameters are obtained by weighted sum of two EMA model parameters. The updated SDP serves as a data prior for distillation during training. After training the model using the sampled batch $(X, Y)$, update the EMA model $f_{\theta_\alpha}$, $f_{\theta_\beta}$.

### A.5 PSEUDOCODE FOR MEASURING KLR AND KGR

Algorithm 2 show how KLR and KGR are measured in practice. We obtain predictions for the test set at two different timestamps, $t_1$ and $t_2$. Then we calculate the joint probability for GT, prediction at $t_1$, and prediction at $t_2$. KL and KG are calculated as the conditional mutual information. Line 6 and line 7 are derived from the equation for the conditional mutual information $I(X; Y|Z) = I(X, Z; Y) - I(Z; Y)$, and equation for computing the mutual information

---

**Algorithm 2** Computing KLR and KGR

---

1: **Input** Test set $(X, Y_{\text{GT}})$, Previous model output $Y_1$, Current model $f_{\theta(t_2)}$, Set of Possible Labels $\mathcal{Y}_{\text{GT}}, \mathcal{Y}_1, \mathcal{Y}_2$

2: $Y_2 = \left[\arg\max f_{\theta(t_2)}(x) \text{ for } x \in X\right]$  ▷ Inference test set with current model

3: **for** $(y_{\text{gt}}, y_1, y_2) \in (\mathcal{Y}_{\text{GT}}, \mathcal{Y}_1, \mathcal{Y}_2)$ **do**

4:      $P(y_{\text{gt}}, y_1, y_2) = \frac{|\{(y_{\text{gt}}, y_1, y_2) \in (Y_{\text{GT}}, Y_1, Y_2)\}|}{|Y_{\text{GT}}|}$  ▷ Calculate joint probability

5: **end for**

6: $KL(t1, t2) = \sum\limits_{y_{\text{gt}} \in \mathcal{Y}_{\text{GT}}} \sum\limits_{y_1 \in \mathcal{Y}_1} \sum\limits_{y_2 \in \mathcal{Y}_2} P(y_{\text{gt}}, y_1, y_2) \log \frac{P(y_1)P(y_{\text{gt}}, y_2)}{P(y_{\text{gt}}, y_1)P(y_1, y_2)}$

                   ▷ Calculate KL by Conditional Mutual Information

7: $KG(t1, t2) = \sum\limits_{y_{\text{gt}} \in \mathcal{Y}_{\text{GT}}} \sum\limits_{y_1 \in \mathcal{Y}_1} \sum\limits_{y_2 \in \mathcal{Y}_2} P(y_{\text{gt}}, y_1, y_2) \log \frac{P(y_2)P(y_{\text{gt}}, y_1)}{P(y_{\text{gt}}, y_2)P(y_1, y_2)}$

                   ▷ Calculate KG by Conditional Mutual Information

8: $TK(t_1) = \sum\limits_{y_{\text{gt}} \in \mathcal{Y}_{\text{GT}}} \sum\limits_{y_1 \in \mathcal{Y}_1} P(y_{\text{gt}}, y_1) \log \frac{P(y_{\text{gt}}, y_1)}{P(y_{\text{gt}})P(y_1)}$

                   ▷ Calculate TK at t1 by Mutual Information

9: $H(Y_{\text{GT}}) = -\sum\limits_{y_{\text{gt}} \in \mathcal{Y}_{\text{GT}}} P(y_{\text{gt}}) \log P(y_{\text{gt}})$  ▷ Calculate the amount of information (entropy) in GT label

10: $KLR(t_1, t_2) = \dfrac{KL(t_1, t_2)}{TK(t_1)}$  ▷ Obtain KLR

11: $KGR(t_1, t_2) = \dfrac{KG(t_1, t_2)}{H(Y_{\text{GT}}) - TK(t_1)}$  ▷ Obtain KGR

12: **Output** $Y_2, KLR(t_1, t_2), KGR(t_1, t_2)$

---

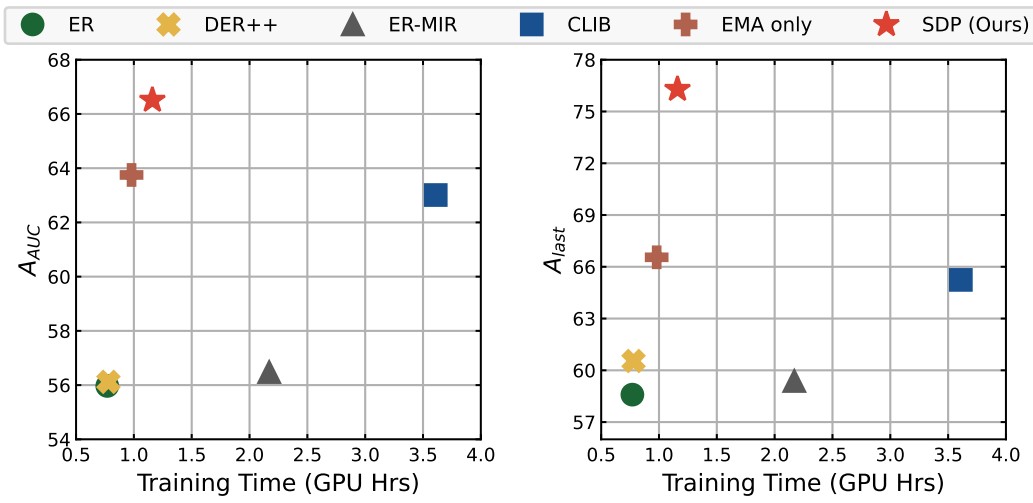

Figure 5: $A_{\text{AUC}}$ and $A_{\text{last}}$ of various methods compared to training time, on Non-Periodic Gaussian data stream in CIFAR-10.

$I(X; Y) = \sum\limits_{y \in \mathcal{Y}} \sum\limits_{x \in \mathcal{X}} P(x, y) \log\left(\frac{P(x, y)}{P(x)P(y)}\right)$. The marginal probabilities are obtained by summation over other dimensions.

## A.6    COMPUTATIONAL COMPLEXITY

We compare the computational complexity by reporting the training time on a single NVIDIA GeForce RTX 2080Ti. 'EMA only' is a method using EMA model for distillation, without us-

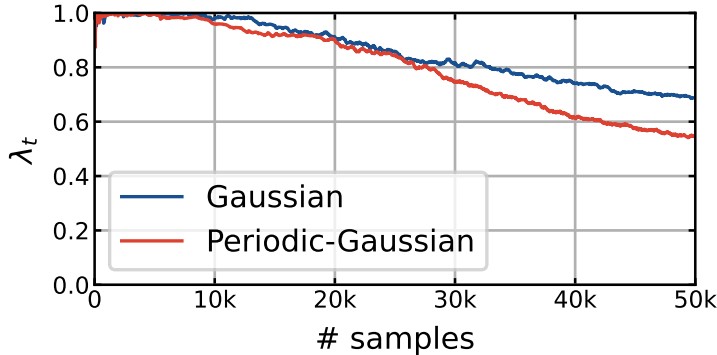

Figure 6: Visualization of balancing parameter $\lambda_t$ over time, on CIFAR-10 Gaussian and Periodic-Gaussian data stream.

ing data-driven loss balancing. We did not report GDumb since GDumb does not train at training time. Instead, GDumb trains from scratch at inference time, so their computational cost depends on the frequency of inference queries. SDP shows good trade-off between computational complexity and performance, compared to other methods.

## A.7 DISCUSSION ON THE ADDITIONAL MEMORY COST

| Method | Additional Memory | |
| --- | --- | --- |
| | Theoretical | ImageNet (MB) |
| ER | $N_s \cdot N_c \cdot S$ | 3,010.6 |
| DER++ | $N_s \cdot N_c \cdot S + N_s \cdot N_c^2$ | 3,090.6 |
| ER-MIR | $N_s \cdot N_c \cdot S + N_\theta$ | 3,057.3 |
| GDumb | $N_s \cdot N_c \cdot S$ | 3,010.6 |
| CLIB | $N_s \cdot N_c \cdot S$ | 3,010.6 |
| SDP (Ours) | $N_s \cdot N_c \cdot S + 2 \cdot N_\theta$ | 3,100.0 |

Table 8: Additional memory cost for storing models, replay buffers, and logits. We report theoretical values and actual memory cost on ImageNet.

We compare the additional memory required to save models or samples in Table. 8. SDP has an additional memory cost of $N_s \cdot N_c \cdot S + 2 \cdot N_\theta$, where $N_s$ is the number of exemplars stored per class, $N_c$ is the number of classes, $S$ is the size of an image, and $N_\theta$ is the size of model parameters. Overall, SDP uses more memory than other compared methods since it requires storing model weights for SDP while other methods only store samples in episodic memory.

Since the resolution of the image in the real world is larger than or similar to the resolution of the ImageNet images, we compare the memory costs in the ImageNet as the closest baseline for the real world applications. As shown in Table. 8, the extra memory cost of storing the model checkpoints is negligible compared to the memory cost of episodic memory.

Storing model parameters from previous tasks for regularization or distillation is a common practice in CL with task boundaries (Kirkpatrick et al., 2017; Chaudhry et al., 2018a; Wu et al., 2019). However, such methods were not actively studied in task-free CL, since one cannot decide which checkpoint to store. For comparison with other regularization or distillation methods, EWC (Kirkpatrick et al., 2017) requires $2 \cdot T \cdot N_\theta$ where $T$ is the number of tasks, EWC++ (Chaudhry et al., 2018a) requires $N_s \cdot N_c \cdot S + 3 \cdot N_\theta$ (3150.6MB on ImageNet), RWalk (Chaudhry et al., 2018a) requires $N_s \cdot N_c \cdot S + 5 \cdot N_\theta$ (3244.1MB on ImageNet), and BiC (Wu et al., 2019) requires $N_s \cdot N_c \cdot S + N_\theta$ (3057.3MB on ImageNet).

## A.8 VISUALIZATION OF LOSS BALANCING PARAMETER OVER TIME

In this section, we discuss the behavior of the data-driven balancing parameter $\lambda_t$ over time. $\lambda_t$ learns to adjust the weight of distillation loss to follow the performance of the SDP model. Thus, in the early phase where SDP model's performance is low, the distillation loss has small weights (high $\lambda_t$), and as the model learns and SDP model's performance increases, the weight for the distillation loss becomes larger (lower $\lambda_t$). We provide visualization of $\lambda_t$ over time in CIFAR-10 Gaussian

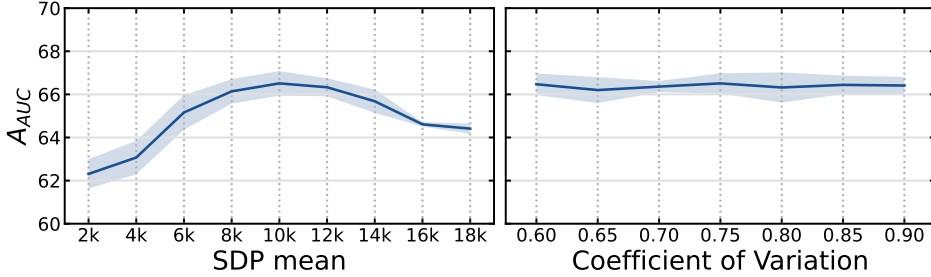

Figure 7: Comparison of different values of mean and coefficient of variation, evaluated on CIFAR-10 non-periodic Gaussian data stream.

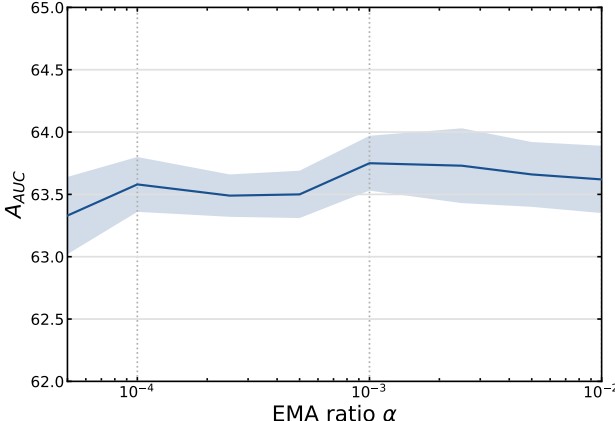

Figure 8: Comparison of different values of EMA ratio for EMA model in Table 1, for CIFAR-10 Gaussian non-periodic Gaussian data stream.

and Periodic Gaussian data stream in Fig. 6, where this behavior is observed. Since the model's predictions are almost random in the initial phase, the coefficient tends to start with 1 and decrease with the training progress.

### A.9 HYPERPARAMETER TUNING ON SDP

We study the effect of two hyperparameters of SDP, mean $\mu$ and coefficient of variation $c^2$, by grid search on CIFAR-10 non-periodic Gaussian setup, and summarize the results in Fig. 7. We observe some dependence on SDP mean, but not much on coefficient of variation. Note that the valid range for coefficient of variation is $0.5 - \frac{1}{\mu} < c^2 < 1 - \frac{1}{\mu}$. Since it is not much relevant to performance, we choose the value in the middle of valid range, $c^2 = 0.75$ for all experiments. SDP mean has some impact on the performance, so we choose the optimal value found on CIFAR-10 non-periodic Gaussian setup, $\mu = 10,000$. Since dataset or setup-specific hyperparameter search is not desirable in CL scenario, we use the same value, $\mu = 10,000$, for all datasets and setups.

### A.10 HYPERPARAMETER TUNING ON EMA

We also report hyperparameter tuning result for EMA model in Table. 1, searched on CIFAR-10 non-periodic Gaussian setup, which is same as the setup on Table. 1. The performance did not vary much depending on the EMA ratio $\alpha$, and we used $\alpha = 0.001$ as it showed highest performance.

### A.11 ONLINE CL WITHOUT REPLAY

We consider an extreme scenario where episodic memory is not available. Since other CL methods covered in this work depend on memory replay, we compare only with basic fine-tuning as baseline. We summarize the result on Table 9. We observed that applying SDP to the baseline (fine-tuning) improves $A_{\mathrm{AUC}}$ by 3.07% for Gaussian data stream and 5.03% for Periodic Gaussian data stream on CIFAR-10. While SDP improves performance, the overall performance falls behind SDP using

| Methods | Gaussian | | Periodic-Gaussian | |
|---|---|---|---|---|
| | $A_{\text{AUC}} \uparrow$ | $A_{last} \uparrow$ | $A_{\text{AUC}} \uparrow$ | $A_{last} \uparrow$ |
| Finetune | 37.78±1.28 | 19.26±1.25 | 31.94±0.85 | 21.99±1.53 |
| SDP (Ours) | **40.85±0.90** | **20.62±1.56** | **36.97±1.19** | **25.35±2.65** |

Table 9: CL with no episodic memory, on Gaussian and Periodic Gaussian data stream in CIFAR-10.

memory replay by 25.66% on Gaussian data stream and 32.70% on Periodic Gaussian data stream. Closing this gap would be an interesting research direction for future work.

## A.12 Ablations on the Number of Repetitions in Periodic Continual Learning

| Methods | $R = 2$ | | | | $R = 3$ | | | |
|---|---|---|---|---|---|---|---|---|
| | $A_{\text{AUC}} \uparrow$ | $A_{last} \uparrow$ | $KLR_{avg} \downarrow$ | $KGR_{avg} \uparrow$ | $A_{\text{AUC}} \uparrow$ | $A_{last} \uparrow$ | $KRL_{avg} \downarrow$ | $KGR_{avg} \uparrow$ |
| ER | 58.51±1.41 | 64.19±5.63 | 22.71±2.64 | 25.00±2.28 | 59.77±0.77 | 63.44±3.13 | 25.67±1.30 | 26.34±1.71 |
| DER++ | 56.68±1.28 | 63.80±4.39 | 22.33±2.11 | 25.95±1.29 | 58.79±1.21 | 64.82±3.18 | 22.21±1.05 | 27.10±2.06 |
| ER-MIR | 58.89±1.04 | 61.87±4.13 | 25.51±2.17 | 24.99±1.76 | 59.92±1.82 | 63.70±2.15 | 24.67±1.16 | 26.56±0.15 |
| GDumb | 44.28±0.20 | 39.87±0.68 | 40.98±2.92 | 11.92±0.47 | 43.62±0.06 | 42.07±0.38 | 37.54±2.23 | 12.48±0.27 |
| CLIB | 64.53±0.06 | 69.60±0.76 | 19.87±0.91 | **26.47±0.10** | 65.41±1.14 | 71.52±0.32 | 19.24±1.08 | **27.64±1.13** |
| SDP (Ours) | **73.01±0.04** | **81.61±0.11** | **3.63±0.56** | 16.88±0.83 | **72.81±0.05** | **81.45±0.25** | **3.71±0.63** | 16.16±0.41 |

| Methods | $R = 4$ | | | | $R = 5$ | | | |
|---|---|---|---|---|---|---|---|---|
| | $A_{\text{AUC}} \uparrow$ | $A_{last} \uparrow$ | $KLR_{avg} \downarrow$ | $KGR_{avg} \uparrow$ | $A_{\text{AUC}} \uparrow$ | $A_{last} \uparrow$ | $KLR_{avg} \downarrow$ | $KGR_{avg} \uparrow$ |
| ER | 60.34±1.91 | 67.84±5.10 | 19.14±0.17 | 25.69±1.86 | 61.17±0.97 | 67.29±2.58 | 17.87±0.48 | 22.44±1.05 |
| DER++ | 58.71±1.10 | 68.77±3.90 | 18.51±0.44 | 26.12±0.95 | 60.70±1.07 | 70.41±1.83 | 16.47±0.53 | 23.45±0.49 |
| ER-MIR | 59.69±2.00 | 64.58±2.58 | 23.01±1.03 | 23.22±1.02 | 60.16±1.70 | 67.86±1.06 | 18.49±1.01 | 22.37±0.91 |
| GDumb | 43.28±0.15 | 42.17±0.52 | 43.86±3.64 | 12.94±0.83 | 43.01±0.45 | 42.97±0.91 | 39.88±1.90 | 12.87±0.83 |
| CLIB | 66.37±0.28 | 72.06±1.05 | 17.45±1.38 | **26.42±0.71** | 66.18±0.64 | 73.86±1.38 | 16.52±0.43 | **24.99±0.69** |
| SDP (Ours) | **72.74±0.09** | **81.54±0.30** | **4.20±0.13** | 17.51±1.31 | **69.68±0.33** | **78.66±0.91** | **5.54±0.22** | 18.54±1.28 |

Table 10: Comparison of Accuracy for different number of repetitions ($R$) for Periodic Gaussian data stream in CIFAR-10

We test CL methods on 4 different number of repetition ($R$) for Gaussian data stream in CIFAR-10 and CIFAR-100. As seen in Table 10, SDP outperform other methods in all tested values of $R$, showing that our model is robust to various number of repetitions.

## A.13 Continual Learning on Gaussian Mixture Data Distribution

In the previous experiments, we used Gaussian distribution for data stream, since data as the sum of independent random variables tend to follow Gaussian distribution due to the Central Limit Theorem. However, a real-world data may not strictly follow Gaussian distribution, so we also consider a more complex distribution than a Gaussian distribution. To test the performance of CL methods on a more complex data distribution, we use Gaussian Mixture, which is a mixture of multiple Gaussian distributions. We model the distribution of each class as the mixture of two Gaussians, where mean for each Gaussian is randomly sampled from $[0, 1)$, standard deviation for each Gaussian is randomly sampled from $[0, 0.2)$, and mixture weight for each class is randomly sampled from $[0, 1)$. Table 11 summarizes the results for Non-Periodic and Periodic Gaussian Mixture data stream. We observe that SDP still outperforms other methods even in a more complex Gaussian Mixture data stream setup.

| Methods | Gaussian Mixture | | | | Periodic Gaussian Mixture | | | |
|---|---|---|---|---|---|---|---|---|
| | $A_{\text{AUC}} \uparrow$ | $A_{last} \uparrow$ | $KLR_{avg} \downarrow$ | $KGR_{avg} \uparrow$ | $A_{\text{AUC}} \uparrow$ | $A_{last} \uparrow$ | $KRL_{avg} \downarrow$ | $KGR_{avg} \uparrow$ |
| ER | 59.89±3.29 | 62.28±3.77 | 22.01±2.57 | 23.51±1.56 | 63.86±0.91 | 68.63±2.53 | 16.82±1.23 | 22.46±0.39 |
| DER++ | 59.06±3.22 | 62.60±5.16 | 21.40±2.93 | **25.29±3.34** | 63.17±0.59 | 68.71±2.17 | 15.64±1.73 | 24.52±1.28 |
| ER-MIR | 57.95±2.77 | 61.01±5.14 | 23.39±4.07 | 23.57±2.96 | 62.71±0.58 | 68.97±1.58 | 17.35±1.61 | 23.64±1.17 |
| GDumb | 43.13±1.71 | 42.42±1.94 | 36.65±3.51 | 11.69±0.31 | 41.61±0.02 | 39.60±0.18 | 48.13±3.45 | 12.73±0.63 |
| CLIB | 65.98±2.24 | 68.81±1.81 | 18.66±1.29 | 24.07±1.43 | 68.00±0.89 | 75.13±2.06 | 15.20±1.95 | **26.01±1.11** |
| SDP (Ours) | **68.85±2.80** | **76.22±0.67** | **4.93±0.26** | 17.63±2.99 | **70.40±1.02** | **79.96±0.40** | **4.19±0.34** | 17.07±1.17 |

Table 11: Comparison of Accuracy for Periodic ($R = 5$) and Non-Periodic Gaussian Mixture data stream in CIFAR-10

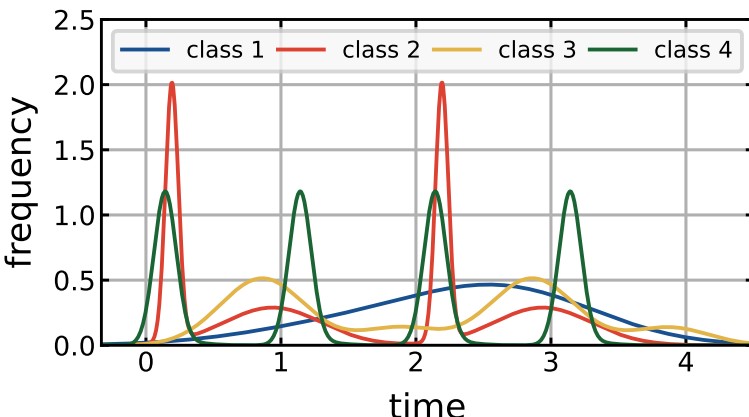

Figure 9: Data distribution of randomly selected 4 classes in the mixed-period Gaussian Mixture data distribution setup.

We also observe in Sec. A.16 that in a real-world data, some classes may have different period with others. To model such scenario, we assign randomly assign period for each class from $1, 2, 4$, where the length of data stream is $4$. For distribution of each class, we use Gaussian Mixture with randomly selected two Gaussians as in the previous experiments. In summary, each class distribution is a periodic two-component Gaussian mixture with randomly selected means, standard deviations, mixture weights, and periods. We visualize an example of such distribution in Fig. 9. We summarize the result on this Gaussian Mixture setup with mixed periods in Table 12. SDP outperforms other methods even on the mixed-period Gaussian Mixture setup with complex data distributions, which indicates that SDP works well on various data distributions.

| Methods | Multi-Periodic Gaussian Mixture | | | |
|---------|---------|---------|---------|---------|
| | $A_{\text{AUC}} \uparrow$ | $A_{\text{last}} \uparrow$ | $KLR_{\text{avg}} \downarrow$ | $KGR_{\text{avg}} \uparrow$ |
| ER | 60.24±0.94 | 66.79±0.63 | 18.83±0.46 | 26.38±0.24 |
| DER++ | 59.97±1.79 | 70.13±0.81 | 17.18±0.84 | **28.35±1.12** |
| ER-MIR | 59.45±1.83 | 67.34±1.25 | 19.73±1.92 | 27.06±1.22 |
| GDumb | 42.35±1.18 | 40.25±1.88 | 37.62±3.54 | 11.60±1.11 |
| CLIB | 66.78±1.12 | 72.96±0.78 | 14.98±0.80 | 26.21±0.98 |
| SDP (Ours) | **69.38±1.21** | **79.20±0.78** | **4.92±0.10** | 18.84±1.00 |

Table 12: Comparison of Accuracy for Periodic Gaussian Mixture data stream with mixed period for each class.

## A.14 ACCURACY TRENDS ON TRAINING PHASE

We visualize the accuracy trends for the tested CL methods in Fig. 10. From the trends, we can observe that SDP performs increasingly well with the training progress, since distillation becomes more important as the amount of knowledge in the past models increase. We also observe that performance in early phases tend to depend on memory management and usage, as methods that use balanced memory and train only on memory, *i.e.* CLIB, EMA, SDP, show much higher accuracy than methods that use reservoir sampling and experience replay, *i.e.* ER, DER, MIR, in the early phases. This is likely because reservoir sampling is highly class-imbalanced in the early phases. As the training progresses, the reservoir memory becomes close to balanced, thus the gap between two memory management closes in the later phase. However, in the mid-to-later phase, the effect of distillation comes into play, and the performance of methods that use balanced memory (CLIB, EMA, SDP) deviates depending on the distillation method used, as SDP ≫ EMA > CLIB (No distillation).

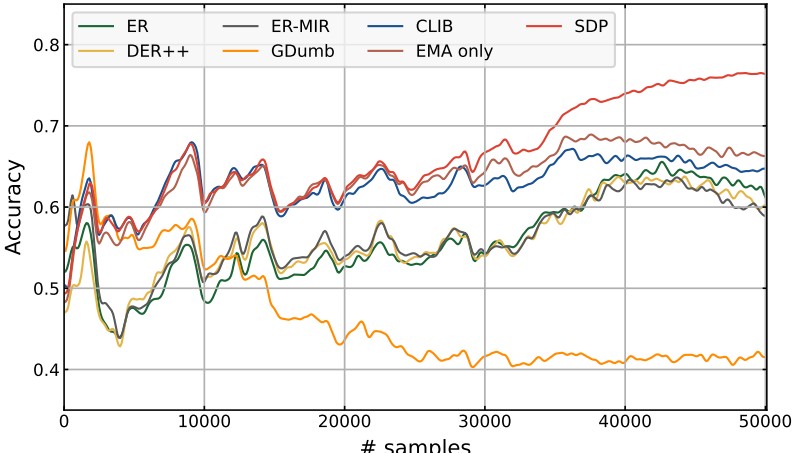

Figure 10: Accuracy trends for CL methods over the course of training, on CIFAR-10 Non-Periodic Gaussian setup.

| Methods | CLEAR-10 | |
| --- | --- | --- |
| | $A_{\text{AUC}} \uparrow$ | $A_{\text{last}} \uparrow$ |
| ER | 44.02±0.84 | 52.91±1.96 |
| DER++ | 38.33±2.09 | 49.82±3.28 |
| ER-MIR | 44.44±1.73 | 56.18±1.32 |
| GDumb | 38.60±0.29 | 40.73±1.71 |
| CLIB | 53.49±0.75 | 66.91±1.27 |
| SDP (Ours) | **56.82±0.35** | **72.97±1.49** |

Table 13: Comparison of Accuracy for Periodic Gaussian data stream with natural domain shift using CLEAR-10.

## A.15 PERIODIC CONTINUAL LEARNING WITH NATURAL DOMAIN SHIFTS

We conduct additional experiments using the recently proposed CLEAR-10 (Lin et al., 2021) dataset that has real-world domain shifts, *i.e.*, the visual concept drift of objects over time in the data stream. It is of particular importance as many real-world systems such as an e-commerce platform would experience gradual domain shifts in the streamed data for the natural temporal evolution of concepts (*e.g.*, computers in 2020 look different from these in 2010), while being subjected to (periodically) changing data distributions. Note that in the original CLEAR-10 benchmark, the data distribution is uniform and stationary. We make a Periodic Gaussian data stream with each bucket as a period and frequency of each class follows a Gaussian distribution within the bucket, as explained in Sec. 3.1. We report the results in Table 13. SDP outperforms other methods by significant margins, even though the Greedy Balancing Sampler we use for memory management is reported as not suitable for domain shifts (Mai et al., 2022). It indicates that the distillation part in SDP can effectively deals with domain shifts in the data stream.

## A.16 EXAMPLES OF REAL-WORLD PERIODIC DATA

In addition to the examples mentioned in Figure 1, many search data follow periodic distribution as shown in Figure 11 and Figure 12. As we can see in Fig. 12, search frequency of seasonal fruits and home appliances, which are greatly affected by the season, follows the periodic distribution with the period of a year. In the case of clothing, the period is 6 months as we can see in Figure 11. It can be seen that almost similar distribution is repeated although there is a slight difference in distribution for each period. In order to address these real-world distributions, we propose the periodic benchmark.

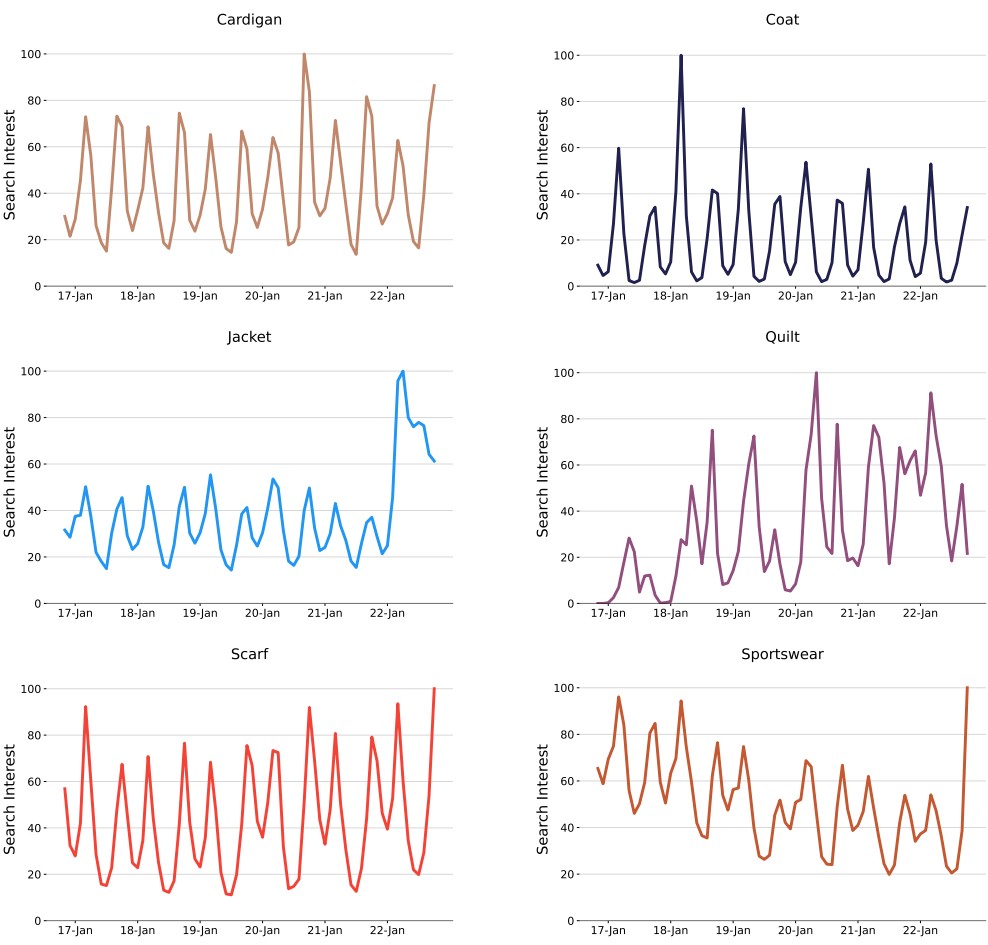

Figure 11: Search data with 6-month period

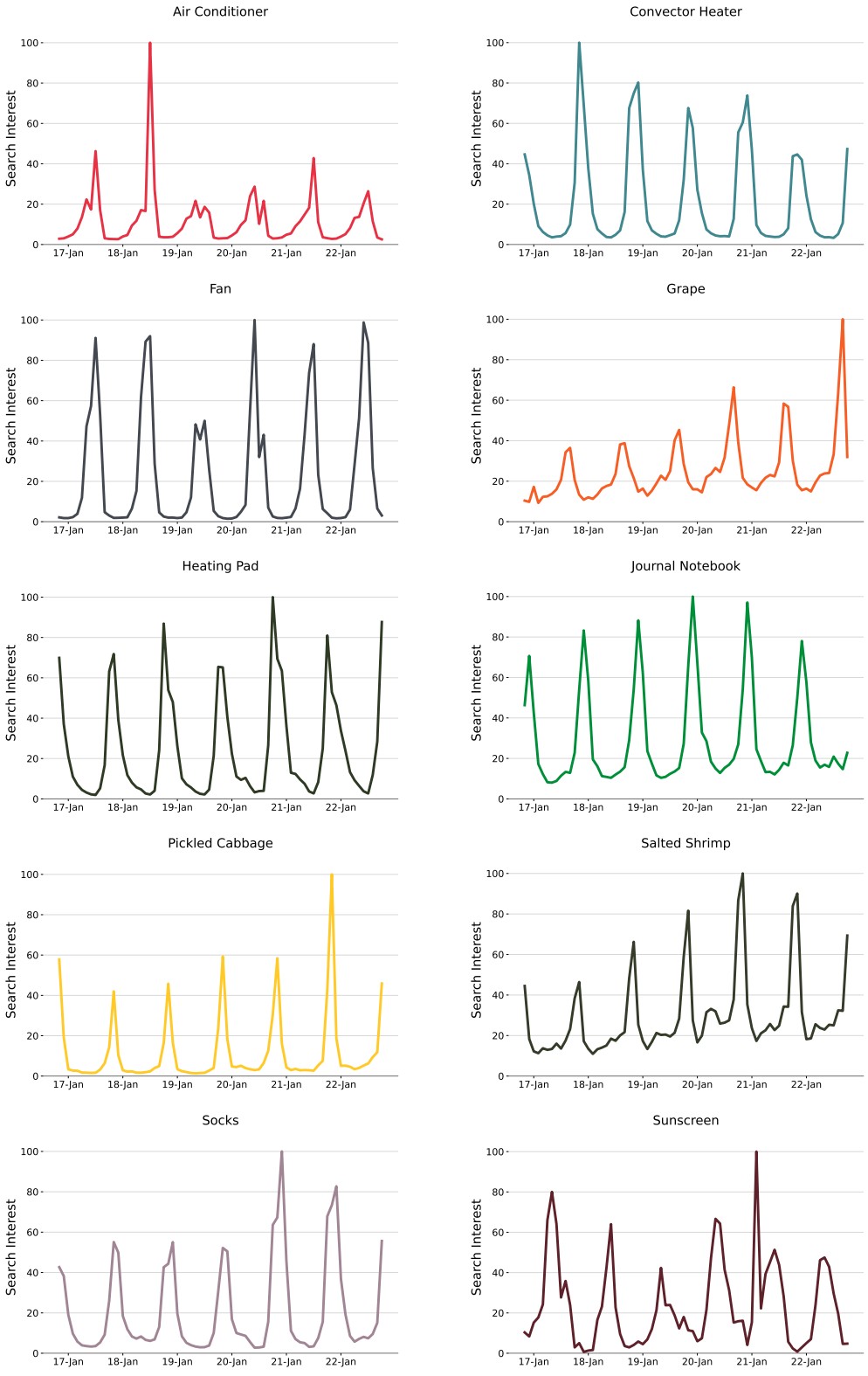

Figure 12: Search data with 1-year period

