# OpenReview forum: "Online Boundary-Free Continual Learning by Scheduled Data Prior"
_ICLR.cc/2023/Conference — ICLR 2023 poster_

### Official Review · Reviewer_bdyM · 2022-10-20

**Confidence:** 3
**Correctness:** 3
**Technical Novelty And Significance:** 3
**Empirical Novelty And Significance:** 3
**Recommendation:** 5

**Clarity, Quality, Novelty And Reproducibility:**

Good quality.
Poor clarity.
Nice originality of the work.

**Strength And Weaknesses:**

My main concern is about the proposed approach.

(1) What is EMA? How to achieve the EMA model ($M_{\alpha}$ or $M_{\beta}$)? How to achieve the distillation model ($f_{SDP}(x)$)? It is necessary to describe them in details.

(2) I as a little bit confused about the core ideal of proposed method. Is the distillation model used to guide training model? Please provide a detail description of your methods by understandable way, like flowchart or pseudocode.

**Summary Of The Paper:**

This work focuses on a more realistic online continual learning---learning continuously changing data distribution without explicit task boundary, which is called boundary-free setup in this paper. The main contributions can be summarized as:

(1) Studying online CL with continuous data stream setup without explicit task boundary, including newly proposed periodic CL setup.

(2) Proposing an online boundary-free CL method that uses scheduled transfer of past knowledge.

(3) Proposing to learn to balance amount of using past and present knowledge.

(4) Proposing new metrics.

**Summary Of The Review:**

Marginally below the acceptance threshold.

---

> ### Author Response · Authors · 2022-11-12
> **Answers to the questions of Reviewer bdyM**
>
> > What is EMA? How to achieve the EMA model ($M_\alpha$ or $M_\beta$)? How to achieve the distillation model ($f_{SDP}(x)$)? It is necessary to describe them in details.
>
> $\to$ An exponential moving average (EMA) is a type of moving average (MA) that places a greater weight and significance on the most recent data points (Grebenkov et al., 2014), which is prevalent in self-supervised learning literature (*e.g.*, BYOL, iBOT, data2vec, etc). It is defined as:
>
> $\theta_\alpha(t) = (1-\alpha) \cdot \theta_\alpha(t-1) + \alpha \cdot \theta(t),$
>
> where $\theta_\alpha(t)$ is the EMA at timestep $t$, $\theta(t)$ is the data point at time $t$, and $\alpha$ is the EMA ratio, a hyperparameter that determines how fast the old data points are decayed. An exponentially weighted moving average reacts more significantly to recent weight changes than a simple moving average (SMA), which applies an equal weight to all observations in the period.
>
> We make the EMA models $M_\alpha$ and $M_\beta$ by calculating EMA for the online model’s parameter with above definition, with two different EMA ratios $\alpha$ and $\beta$. In the revision, we change the notation from $M_\alpha$ to $\theta_\alpha$, to imply that EMA is calculated with the model's parameters.
>
> **The distillation model ($f_{SDP}(x)$)**: It is a neural network with parameters $\theta_{SDP}$ obtained by eq.(8), as the weighted sum of the two EMA model parameters $\theta_\alpha$ and  $\theta_\beta$
>
> We add the definition of EMA and further details of EMA and SDP models in Sec.5 and Sec.A.4 of the revision. Thank you!
>
>
> > The core ideal of proposed method. Is the distillation model used to guide training model?
>
> $\to$ Yes, the distillation model guides the training model to mitigate the forgetting of the knowledge learned in the past while learning the current data. According to the taxonomy of continual learning methods summarized in (De Lange et al., 2021), it can be categorized as a ‘data-focused method’ in the ‘regularization based continual learning method’ group.
>
> > > Please provide a detail description of your methods by understandable way, like flowchart or pseudocode
>
> $\to$ Great suggestion! We add a pseudocode in Sec.A.4 of the revision. If it is still not clear, please let us know.

---

> ### Author Response · Authors · 2022-11-18
> **Discussion reminder**
>
> We sincerely thank you for your effort in reviewing our submission. We gently remind the reviewer that we tried our best to address your concerns via our replies and revision of the manuscript. As the discussion period is nearing the end, we would be delighted to hear more from you if there are any further concerns.

---

### Official Review · Reviewer_Y7gj · 2022-10-25

**Confidence:** 4
**Correctness:** 3
**Technical Novelty And Significance:** 3
**Empirical Novelty And Significance:** 3
**Recommendation:** 6

**Clarity, Quality, Novelty And Reproducibility:**

The overall quality of the paper are good, despite that some parts are not clear.
The topic and settings are novel.
The results should be reproducible with details in the paper and released code.

**Strength And Weaknesses:**

Strengths:
- The topic of boundary-free CL is interesting and important.
- The proposed metrics to measure loss in boundary-free CL is novel.
- The data-driven balancing part is well-motivated

Weakness:
- The description of the EMA part is a bit unclear.
- I am not very clear about why periodic Gaussian distribution is actually challenging in the CL setting. The periodic natural of the distribution naturally serves as a "replay" mechanism during the CL process. Therefore, shouldn't the setting be easier than the existing Gaussian schedule setting?

**Summary Of The Paper:**

The paper presents a novel online boundary-free CL method on the realistic boundary-free CL setting, including the newly proposed periodic setup. Moreover, new mutual information-based metric is proposed to measure the loss of past knowledge and gain of new knowledge. Comprehensive experiments have been conducted to evaluate the effectiveness of the method.

**Summary Of The Review:**

I think the paper focus on an important problem, boundary-free CL, with a relatively novel solution. Therefore, I recommend weak acceptance initially.

---

> ### Author Response · Authors · 2022-11-12
> **Answers to the questions of Reviewer Y7gj**
>
> > The description of the EMA part is a bit unclear.
>
> $\to$ The exponential moving average (EMA) is a type of moving average (MA) that places a greater weight (*i.e.*, significance) on the most recent data points (Grebenkov et al., 2014). It is defined as $\theta_{\alpha(t)} = (1- \alpha ) \cdot \theta_{\alpha(t-1)} + \alpha \cdot \theta(t)$, where $\theta_\alpha(t)$ is EMA at timestep $t$, $\theta(t)$ is the datapoint at timestep $t$, and $\alpha$ is the EMA ratio. An exponentially weighted moving average reacts more significantly to recent weight changes than a simple moving average (SMA), which applies an equal weight to all observations in the period. We clarify the description in Sec. 5.1 of the revision.
>
>
> > Why periodic Gaussian distribution is actually challenging in the CL setting. The periodic natural of the distribution naturally serves as a "replay" mechanism during the CL process. Therefore, shouldn't the setting be easier than the existing Gaussian schedule setting?
>
> $\to$ Great point! Yes, we agree that the periodic Gaussian may be easier in terms of forgetting as the data distribution has a certain repetition. But note that the repetition in the data distribution does not imply the “replay” of the same data since in the peaks of the distribution, the exact same data would not appear again but the data in the same class would appear again.
>
> Although the periodic Gaussian distribution may not be more challenging, it is the scenario that may often appear in real-world data (as shown in Figure 1 and we also provide further examples of periodic real-world data in Sec. A.12 of the revision). Since this setup is realistic and practical, we believe our setup can give insights to future CL research even if it incurs less forgetting (the similar trend has been observed in the blurry setup (Aljundi et al., 2019c; Prabhu et al., 2020; Bang et al., 2021)). Moreover, existing CL methods may differently perform in the periodic setup than in non-periodic setup, due to the different trade-off between stability and plasticity required for the two setups. For example, methods specialized in preventing forgetting may be less effective in periodic setup, since repetition in distribution that incurs alleviate forgetting would dilute the superiority of that method. Therefore we respectfully argue that it is important to evaluate CL methods in the periodic scenarios for developing an effective CL system for real-world applications.

---

> ### Author Response · Authors · 2022-11-18
> **Discussion reminder**
>
> We sincerely thank you for your effort in reviewing our submission. We gently remind the reviewer that we tried our best to address your concerns via our replies and revision of the manuscript. As the discussion period is nearing the end, we would be delighted to hear more from you if there are any further concerns.

---

### Official Review · Reviewer_1ivp · 2022-10-27

**Confidence:** 4
**Correctness:** 4
**Technical Novelty And Significance:** 4
**Empirical Novelty And Significance:** 4
**Recommendation:** 8

**Clarity, Quality, Novelty And Reproducibility:**

This paper addressed a more realistic and challenging continual learning setup where the task boundaries are unknown. It also proposed a novel evaluation metric to identify the knowledge loss and knowledge gain for previous and current tasks. The paper includes a solid ablation analysis to support the claims.

**Strength And Weaknesses:**

Strengths:
1- Most of the existing continual learning approaches have proposed the task-split setup in which task boundaries are available and well exploited
in various offline/online CL methods. However, the task-split setup is not suitable for the realistic scenario. In contrast to the prior work, this paper proposes a boundary-free task setup that is more challenging and aligned with the realistic scenario. It removes the notion of artificial task boundary; instead, the data arrival follows a certain distribution.

2- The conventional approaches require the accuracy of the previous task to measure the knowledge forgetting, which is not possible in the proposed boundary-free setup. Therefore this paper presents new metrics for measuring forgetting and the ability to learn new knowledge by computing loss and gain of knowledge based on information theory.

3- To determine the amount of past knowledge to be transferred in a continuous fashion, it proposes a method to leverage the previously learned knowledge by a skewed bell-shaped weighting function in a distillation framework named scheduled data prior.

4- The proposed method shows significant improvement in all the benchmark datasets compared to the state-of-the-art techniques for both the setup, task boundary-free, and disjoint task split.

5-  A detailed ablation study is provided to support the significance of the proposed approach.

 Weaknesses:
1-  It assumes a strict assumption that the streaming data comes from the gaussian distribution, which is not practically true because real data follows a complex distribution. Is it possible to predict the task boundaries based on identifying out-of-distribution distribution between tasks? Or it would be more interesting if it covered complex distribution.

2-  The extreme case for continual learning is where the data examples are available only once, unlike the replay-based approaches, where the old data may be available for future training. Applying the proposed framework to this challenging continual setup would be interesting.











**Summary Of The Paper:**

This paper proposes a more challenging and realistic setup for online- continual learning in which there is no explicit task boundary called a boundary-free setup. Due to the lack of task boundaries, it is difficult to find past information that needs to be preserved to identify the stability-plasticity dilemma. It proposes a scheduled transfer of previously learned knowledge to address this issue. It also suggests learning to balance the amount of using past and present knowledge. Further,  this paper proposes new metrics to measure the loss of past knowledge (i.e., forgetting) and the gain of new knowledge based on information theory.










**Summary Of The Review:**

Overall, I found the paper interesting and conducted sufficient experiments for justifications. The paper addresses more challenging and practical issues in the existing continual learning approaches. For more details, please see the strength and weaknesses section.

---

> ### Author Response · Authors · 2022-11-12
> **Answers to the questions of Reviewer 1ivp**
>
> > A strict assumption that the streaming data comes from the gaussian distribution, which is not practically true because real data follows a complex distribution. Is it possible to predict the task boundaries based on identifying out-of-distribution distribution between tasks?
>
> $\to$ Great point! We also do not believe that the Gaussian is the true data distribution. However, we chose it as one of the possible distributions based on the central limit theorem since many random variables (*i.e.*, the factors affecting the data distributions) would be summed up to configure a real-world data distribution.
>
> It is very interesting to utilize out-of-distribution detection methods to identify implicit task boundaries. Given that there are blurry task boundary setup (Prabhu et al., 2020) and its variants (Koh et al., 2022), we expect that detecting the implicit task boundaries is not as trivial as we expect. We think it is a great future research avenue that we would like to pursue. Thank you!
>
> > > Or it would be more interesting if it covered complex distribution.
>
> $\to$ Great suggestions! We are also curious about addressing more complex distributions. As one of the candidate distributions, we consider the mixture of Gaussian to evaluate our proposed SDP and some relevant prior arts. In the following revision, we will add the results and discuss the behavior of the continual learning methods. It would be greatly appreciated if you share your thoughts on the candidate distributions.
>
>
> > The extreme case for continual learning is where the data examples are available only once, unlike the replay-based approaches, where the old data may be available for future training. Applying the proposed framework to this challenging continual setup would be interesting.
>
> $\to$ Great suggestion again! We are also very intrigued about the performance of continual learning methods without the episodic memory replay. For this end, we conduct an additional experiment by applying SDP to online CL with no memory replay. We observed that applying SDP to the baseline (fine-tuning) improves $A_\text{AUC}$ by $3.07\%$ for the non-periodic Gaussian data stream and $5.03\%$ for the periodic Gaussian data stream on CIFAR-10. We were not able to compare with other methods since they all depend on memory replay (and removing the memory makes the method half complete). While our SDP improves performance, the overall performance falls behind the version using memory replay. It would be an interesting direction for future work to close this gap. We summarize the results in Sec. A.11 of revision.

---

> > ### Comment · Reviewer_1ivp · 2022-12-11
> > **Post Rebuttal**
> >
> > Thanks to the authors for their responses and clarifications. After reviewing other reviews and authors' responses, I feel that the paper has a sufficient contribution and qualifies my threshold for acceptance.

---

### Official Review · Reviewer_46rh · 2022-10-31

**Confidence:** 4
**Correctness:** 3
**Technical Novelty And Significance:** 2
**Empirical Novelty And Significance:** 2
**Recommendation:** 6

**Clarity, Quality, Novelty And Reproducibility:**

**Clarity**: In general, this paper is easy to follow, but there are quite some places which could be elaborated on:
- it is not too clear how KLR/KGL are estimated in practice. If I understood this correctly, then in practice, the authors compute the accuracies/probabilities for some set of classes at different timestamps and then those accuracies/probabilities are then plugged in into the equations from Section 4 (Please, correct me if I am wrong). Ideally, a pseudo-code would help a reader to understand this better.
- it is not immediately clear from what is the difference between "boundary-free" and just traditional task-free CL and I didn't find it to be discussed in the related work section.
- it is not quite clear why "scheduled data prior" is called so. If I understand it correctly, there is no schedule in the models' mixing strategy, it is just an analog of EMA.
- there are no details on the experiments from Table 1 (how many tasks, whether it is disjoint or not etc.)
- it is not specified whether a replay buffer is used to rehearse the previous knowledge (like in DER) or not (like in LwF)

**Quality**. Though, the paper would benefit from additional visualizations and ablations. For example, visualize how the loss balancing parameter change over time. Or show how the performance changes depending on the number of tasks. Also, it would be very interesting to see how much memory is being used during training (for storing the teachers and/or the replay buffer) compared to other methods. Also, the comparison to EMA was performed without tuning EMA hyperparameters.

**Novelty**. All 4 contributions are novel, but seem minor or not too beneficial.

**Reproducibility**. I believe that the paper contains enough details to be reproduced (up to some negligible differences). Also, the authors aim to release the source code.

**Strength And Weaknesses:**

Strengths:
1. The overall approach is more or less simple.
2. Authors explore more realistic setups of CL, which feels to me being a big problem.
3. The experimental evaluation looks quite comprehensive.
4. The writing is easy to follow.

Weaknesses:
1. While it's true that there is a lot of real-world data is periodic, I do not understand how improving on the proposed periodic benchmarks can improve the performance of some real-world system which operates on periodic data. Could you please provide me an example of some real-world system which operates on periodic data and suffers from forgetting?
2. In terms of complexity/performance trade-off, just using EMA without learning the balancing coefficient looks better to me.
3. The proposed metrics:
    - are not well-motivated. It is not clear why $A_\text{AUC}$ is bad, what important properties do KLR and KGL measure? Why should we measure KLR and KGL, they seem to always correlate with $A_\text{AUC}$ anyway. For (2),
    - seem to be overly-complicated. If I understood Section 4 correctly, at the end those information-theory-based metrics go down to "average accuracy loss over some set of classes" or "average accuracy gain over some set of classes". Then, why not measure these values directly? It would be both more intuitive and have better scaling (one always know the lower/upper bound for them).
4. Learning the balancing coefficient looks tricky to me. The model can just learn to ignore rehearsing the past data completely. Could you please provide the plots of how this balancing coefficient changes? Also, how do you initialize it? Do you regularize it somehow?

Questions:
1. Do you use any replay buffer for the knowledge distillation loss or you compute it on the currently available data?
2. Do I get it right, that for EMA you store just a single checkpoint of the past model, and for SDP you store two checkpoints?


**Summary Of The Paper:**

This paper studies online continual learning in the boundary-free setting, i.e. we do not know when a task starts or ends (or whether it ends at all, like in the periodic setup). The paper then proposes two metrics: knowledge loss and knowledge gain, which measure the difference in performance between two time points t_1 and t_2 (i.e., how the performance improves/worsens between $t_1$ and $t_2$).

To prevent forgetting, the paper builds upon knowledge distillation strategies. To accumulate the previous weights for the teacher model, the authors propose a different schedule, which they call Scheduled Data Prior (SDP) and show that it works better than EMA. They also use a learnable coefficient to balance between the current classification loss and the knowledge distillation loss.

The ideas are tested on CIFAR 10/100 and TinyImageNet/ImageNet and the proposed approach beats the previous SotA (i.e., CLIB) by ~5-10% on them in terms of $A_\text{AUC}$ on all the setups.


**Summary Of The Review:**

This paper feels like a "big bag of small contributions", where each of the contributions could be dropped without affecting the overall paper quality much. In terms of complexity/performance tradeoff, just using EMA without the learnable loss balancing coefficient still looks better to me (i.e. it would be my way to go if I would start the project on CL right after reading the paper).

The strongest part of this paper is the scores on the leaderboard, but I **subjectively** believe that just the good leaderboard scores shouldn't be a reason for acceptance for a continual learning paper, because the modern CL benchmarks are speculative and too far from the real world anyway, and the improved performance on them does not translate into a clear practical benefit of some real-world system (in contrast to, e.g., modern object detection or segmentation). I am ready to discuss this latter statement and change my mind.

---

> ### Author Response · Authors · 2022-11-12
> **Answers to the questions of Reviewer 46rh (4/4)**
>
> > Show how the performance changes depending on the number of tasks.
>
> $\to$ As the number of tasks is not defined in our “boundary-free” setup, we are not able to measure the performance trend as a function of the number of tasks. But if we consider the number of repetitions as an alternative measure for the number of tasks in our periodic setup, we can report the suggested performance. We are working on plotting the performance when the number of repetitions changes and will report the results in the following revision.
>
>
> > it would be very interesting to see how much memory is being used during training (for storing the teachers and/or the replay buffer) compared to other methods.
>
> $\to$ SDP uses 3,100MB, while the other methods use 3,011MB~3,091MB for storing models and buffers during training on ImageNet. We respectfully argue that this memory increase is negligible, considering the significant performance gain by the SDP against other baselines. We add this in Sec.A.7 of the revision. Thank you!
>
>
> > The comparison to EMA was performed without tuning EMA hyperparameters.
>
> $\to$ We searched its hyperparameters and chose $\alpha=0.001$ for its best empirical performance. We include this in Sec. A.10 of the revision.
>
>
> > “Big bag of small contributions,” where each of the contributions could be dropped without affecting the overall paper quality much.
>
> $\to$ We respectfully argue that all contributions are proposed towards a single objective of addressing a more realistic continual learning setup that has no explicit task boundary for the following reasons:
>
> - Since the setup is new, the previously proposed evaluation metric is not applicable (Sec. 4). Thus, to properly evaluate the methods in this setup, we propose a new set of metrics and a method.
>
> - Since the setup is new, the prior arts are not expected to perform well since they do not consider the more realistic constraints of the new setup (*e.g.*, no task boundary and periodic data distribution). Thus, we propose a simple approach (as a reference for future research) to address this new task setup.
>
>
> > In terms of complexity/performance tradeoff, just using EMA without the learnable loss balancing coefficient still looks better to me (i.e. it would be my way to go if I would start the project on CL right after reading the paper).
>
> $\to$ Thank you for the interesting opinion. We implement the suggested method, compare it with our method, and summarize the results in Fig. 5 in Sec. A.6 of the revision. As shown in the figure, the proposed SDP shows noticeable gain in performance ($A_\text{last}$:+9.74% $A_\text{AUC}$:+2.76%) with small increase in computational complexity (x1.18) compared to using EMA. We cordially argue that the proposed SDP is in better complexity/performance tradeoff for its noticeable accuracy gain at the expense of marginal computational complexity increases.
>
>
> > Improved performance on leaderboard does not translate into a clear practical benefit of some real-world system (in contrast to, e.g., modern object detection or segmentation). I am ready to discuss.
>
> $\to$ We completely agree that the improving accuracy on the existing benchmarks may not translate to practical benefit as the setup that the improvements are achieved may be far from the real-world scenarios. Rather than simply aiming to improve performance in less realistic CL setups, in this work, we speculate how models would receive information from the online data stream, alongside other prior arts that try to propose more realistic scenarios for continual learning (Aljundi et al., 2019c; Prabhu et al., 2020; Koh et al., 2021). Discussing the practicality for the real-world system is the primary contribution of this work.

---

> > ### Comment · Reviewer_46rh · 2022-11-15
> > **Reviewer's response**
> >
> > Thank you a lot for your comprehensive response. I have several concerns now, but several concerns still remain:
> > - *About the benchmark being speculative (partially resolved)*. Your provided example of the e-commerce store made your benchmark make more sense to me, thank you for this. But I still have the concern about why improving on your benchmark could improve that e-commerce system: it's underlying data distribution is likely different, periodicity pattern is different, training objectives are likely different. What would resolve such a concern for me is testing on such kind of a real-world setup directly, instead of emulating it with CIFAR/ImageNet (but I understand how problematic it is).
> > - *About the metrics (resolved)*. Your explanation on the KGR/KLR metrics have resolved my concerns about them.
> > - *About SDP vs EMA (unresolved)*. This is subjective, but the 2.76% gain does not feel worth using SDP. I do not really know what to do with this concern, maybe we should go to the AC with it. I subjectively think that the complexity/improvement tradeoff is not worth it, you subjectively think that it is worth it... I've also just found two CL works which use this EMA idea for knowledge distillation: [[1]](https://openaccess.thecvf.com/content/CVPR2022/papers/Simon_On_Generalizing_Beyond_Domains_in_Cross-Domain_Continual_Learning_CVPR_2022_paper.pdf), [[2]](https://openreview.net/pdf?id=uxxFrDwrE7Y). Do I get it right that it is not possible to treat that EMA model as your paper's contribution because of that?
> >
> > Also, I have some other questions:
> > - *About the balancing parameter*. Thank you for the clarification and I am sorry for the misunderstanding. Could you please tell what subset of images do you use to compute it? Replay buffer? All the images in the dataset?
> > - *For the memory stored during training*. I'm sorry I'have very poorly formulated my question. What I meant is the amount of memory not in terms of the occupied disk space (since it is too dependent on the underlying data type), but in terms of "asymptotics". Could you report the results in terms of "number of examples stored + number of model weights (or equivalents — e.g., gradients) stored". For example, EWA would have the formula of "0 + $T \cdot N_\theta$", where $T$ is the number of tasks (they store gradients per task, right?), $N_\theta$ is the number of parameters. For simple empirical replay (if we store equal amount of examples per class), it would be "$N_s \cdot N_c + 0$", where $N_s$ is the number of examples stored per class, $N_c$ is the number of classes. For SDP, as far as I understand it now, it is "$N_s \cdot N_c + 2 \cdot N_\theta$", right?
> >
> > I have increased my score to 6, since some of my concerns have been resolved and since the largest part of my concern is just my *subjective* negative attitude towards the modern CL benchmarks (I believe they are just way too speculative): it would be unfair if my subjective attitude will affect your submission that much when the rest of the community feels fine about them. Also, maybe your work would nudge future research towards making CL more realistic and grounded.
> >
> > I am ready to increase my score further, if you would perform comprehensive evaluation on some "significantly more real-world-y" benchmark.
> >
> > I apologize that some of my questions were stupid and if some of my statements could sound ill-mannered. Thank you once again for your response.

---

> > > ### Author Response · Authors · 2022-11-17
> > > **Author’s response to the reviewer 46rh’s response (2/2)**
> > >
> > >
> > > > it would be unfair if my subjective attitude will affect your submission that much when the rest of the community feels fine about them. Also, maybe your work would nudge future research towards making CL more realistic and grounded. I apologize that some of my questions were stupid and if some of my statements could sound ill-mannered. Thank you once again for your response.
> > >
> > > $\to$ We sincerely appreciate your detailed concerns and careful comments on our submission. We particularly appreciate the diverse opinions and concerns for the proposed setup, which we believe to make research reliable by addressing the concerns. As there are many continual learning method proposals in recent literature, as you also mentioned, we also deeply believe the unrealistic setups that have been used to evaluate the methods make the research progress less meaningful. In light of these thoughts, we try to give a nudge to alert the community to reconsider the setups we might have taken for granted; that is the primary purpose of this work. We again appreciate your comments and we are encouraged to move forward in addressing more realistic CL.

---

> > > ### Author Response · Authors · 2022-11-17
> > > **Author’s response to the reviewer 46rh’s response (1/2)**
> > >
> > > > (*About the benchmark being speculative*) I still have the concern about why improving on your benchmark could improve that e-commerce system: its underlying data distribution is likely different, periodicity pattern is different, training objectives are likely different. What would resolve such a concern for me is testing on such kind of a real-world setup directly, instead of emulating it with CIFAR/ImageNet (but I understand how problematic it is).
> > >
> > > $\to$ We totally agree on the importance of evaluating methods on a real-world data stream for the reasons you have raised and eagerly wanted to evaluate the methods on real data such as the e-commerce data. However, as the problem of using the real data including various legal issues and technical limitations such as leaking private information and filtering inappropriate data still persists, we have not been allowed to access the real data. Instead, we conduct additional experiments to take a few steps closer to the real world set up by using the following distributions:
> > > For the periodic/non-periodic data, the data distribution follows the Gaussian Mixture Model (Table 11 in Sec. A.14 of the second revision).
> > > For the periodic data, the period of distribution is different for each class. To be more specific, the entire data stream was split at random periods (Table 12 in Sec A.14 of the second revision), which resembles the period of the e-commerce service data we presented. (Please refer to Fig. 11 in Sec A.14 of the second revision for the visualization of this distribution)
> > >
> > > As shown in the results, we observe that the SDP still performs well for the seemingly more realistic scenarios since the SDP does not assume a specific data distribution (*e.g.*, Gaussian). Although this is not the empirical evidence for the *real* data, it would be worthwhile to report as the distribution is reasonably similar to the real world one (please compare Fig. 1 and Fig. 11 for the similarity).
> > >
> > >
> > >
> > > > (*About SDP vs EMA*) This is subjective, but the 2.76% gain does not feel worth using SDP. I do not really know what to do with this concern, maybe we should go to the AC with it.
> > >
> > > $\to$ In $A_{auc}$, the performance difference between EMA and SDP may look marginal (+2.76%), but in $A_{last}$, the improvement is noticeably large (+9.74%) (refer to Fig. 5 in Sec. A.6). Given that $A_{last}$ is a widely used metric, we respectfully argue that the SDP exhibits noticeable empirical gain over the EMA. We could explain the relatively smaller gain in $A_{auc}$ as follows. It is because the choice of ‘data prior’ (*i.e.*, either EMA or SDP) does not much affect the accuracy in the early learning phase, since there is little knowledge contained in the past models. However, as the learning progresses and the past models contain important knowledge about the past, the choice of data prior becomes important, and SDP performs increasingly better than EMA, up to +9.74% in the final accuracy (Fig. 12 in Sec. A.15 of the revision).
> > >
> > > > > I subjectively think that the complexity/improvement tradeoff is not worth it, you subjectively think that it is worth it.
> > >
> > > $\to$ We appreciate your concern about this issue. We believe that the proposed method is beneficial in the tradeoff; the SDP only costs $1.18\times$ computation of the EMA in CIFAR-10 but achieves $+9.74\%$ improvement in $A_{last}$. Note that computational overheads by CLIB and MIR are larger than this (CLIB: $3.68\times$, MIR: $2.21\times$) with worse performance than EMA.
> > >
> > > > > I've also just found two CL works which use this EMA idea for knowledge distillation: [1], [2]. Do I get it right that it is not possible to treat that EMA model as your paper's contribution because of that?
> > >
> > > $\to$ Yes, you are correct!
> > >
> > >
> > > > (*About the balancing parameter*) Could you please tell what subset of images do you use to compute the balancing parameter? Replay buffer? All the images in the dataset?
> > >
> > > $\to$ We use the latest 50 images per class in the data stream to compute the balancing parameter. Note that we only have to save the confidence of each sample (*i.e.*, a single number per sample), not images, to calculate the balancing parameters.
> > >
> > > > (*For the memory stored during training*) The amount of memory in terms of "asymptotics". Could you report the results in terms of "number of examples stored + number of model weights (or equivalents — e.g., gradients) stored". For SDP, as far as I understand it now, it is "$N_s \cdot N_c + 2 \cdot N_\theta$", right?
> > >
> > > $\to$ Yes, you are correct! SDP has a memory cost of $N_s \cdot N_c + 2 \cdot N_\theta$. We add the memory cost in terms of asymptotics in Table. 8 of Sec. A.7 in the revision.

---

> > > ### Author Response · Authors · 2022-11-18
> > > **Additional response by the authors to the reviewer 46rh’s response**
> > >
> > > Taking your concerns on real-world performance of CL methods seriously, besides the experiments on Gaussian mixture distributions with different periods (as a similar distribution to the real world data), we conduct additional experiments using the recently proposed CLEAR-10 (Lin et al., 2021) dataset that has real-world *domain shifts*, *i.e.*, the visual concept drift of objects over time in the data stream. It is of particular importance as many real-world systems including an e-commerce platform would experience gradual domain shifts in the streamed data for the natural temporal evolution of concepts (*e.g.*, computers in 2020 look different from these in 2010), while being subjected to (periodically) changing data distributions. Note that in the original CLEAR-10 benchmark, the data distribution is uniform and stationary. Particularly in our experiments, we use the periodic Gaussian data distribution of CLEAR-10 data, evaluate all the CL methods we have compared in the tables in the submission (ER, DER++, ER-MIR, CLIB, GDumb and SDP), and summarize the results in Table 13 in Sec. A.16 of the third revision. As shown in the table, SDP again outperforms other methods by significant margins.
> > >
> > > **Bibliography**
> > >
> > > Zhiqiu Lin, Jia Shi, Deepak Pathak, and Deva Ramanan. The CLEAR benchmark: Continual learning on real-world imagery. In NeurIPS Datasets and Benchmarks Track (Round 2), 2021.

---

> ### Author Response · Authors · 2022-11-12
> **Answers to the questions of Reviewer 46rh (3/4)**
>
> > Learning the balancing coefficient looks tricky to me. The model can just learn to ignore rehearsing the past data completely. Could you please provide the plots of how this balancing coefficient changes?
>
> $\to$ Good suggestion! We add the plot of the balancing coefficient over time in Sec. A.8 (Fig. 6) of the revision. Note that the balancing coefficient is not optimized to minimize a loss, but rather to adjust the weight of distillation loss to follow the performance of the SDP model. Thus, as shown in the figure, in the early phase where the SDP model's performance is low, the distillation loss has small weights (high $\lambda_t$). As the model learns and the SDP model performs better, the weight for the distillation loss becomes larger (lower $\lambda_t$).
>
> > > Also, how do you initialize it?
>
> $\to$ We initialize the coefficient by Eq.(12).
>
> > > Do you regularize it somehow?
>
> $\to$ No.
>
>
> > Do you use any replay buffer for the knowledge distillation loss or you compute it on the currently available data?
>
> $\to$ We use a replay buffer for knowledge distillation loss, following (Wu et al., 2019; Buzzega et al., 2020). Please refer to Sec. 6 for more details.
>
>
> > For EMA, do you store just a single checkpoint of the past model, and for SDP, do you store two checkpoints?
>
> $\to$ Yes, both are correct.
>
>
> > It is not too clear how KLR/KGL are estimated in practice. If I understood this correctly, then in practice, the authors compute the accuracies/probabilities for some set of classes at different timestamps and then those accuracies/probabilities are then plugged in into the equations from Section 4.
>
> $\to$ You’re correct -- note that we only use the probabilities and not the accuracies. In more detail, we obtain predictions for the test set at two different timestamps, $t_1$ and $t_2$. Then we calculate the joint probability and marginal probabilities for GT, prediction at $t_1$, and prediction at $t_2$. We plug the probabilities into the equations in Sec. 4.
>
> > > Ideally, a pseudo-code would help a reader to understand this better.
>
> $\to$ Great suggestion! We add the pseudocode to the Sec. A.5 in the revision. Thank you.
>
>
> > It is not immediately clear from what is the difference between "boundary-free" and just traditional task-free CL and I didn't find it to be discussed in the related work section.
>
> $\to$ The difference is that in the traditional task-free set-up, the distribution of incoming data itself is divided into explicit tasks but the model does *not use information about the tasks* during training.  In other words, the data distribution in the traditional task-free set-up is fixed per task. In contrast, in the proposed “boundary-free”, the data is sampled from a continuous distribution over time with no notion of task boundaries. We have discussed this in Sec. 1 and now add the discussion to the related work (Sec. 2) as well in the revision. Thank you!
>
>
> > It is not quite clear why "scheduled data prior" is called so. If I understand it correctly, there is no schedule in the models' mixing strategy, it is just an analog of EMA.
>
> $\to$ We used the term 'scheduled' since the amount of weight is a function of *time*. (In this sense, EMA also can be considered as a schedule) Unlike the EMA’s simple monotonically decreasing schedule, we propose to consider that knowledge transfer should focus more on past knowledge than recently learned one, which leads to empirical benefits (Tab. 1 in Sec. 5.1).
>
>
> > There are no details on the experiments from Table 1 (how many tasks, whether it is disjoint or not etc.)
>
> $\to$ Unlike conventional CL setups, the number of tasks is not defined thus the task configuration is neither disjoint nor blurry in our proposed setup as the data is sampled from a continuous data distribution (*e.g.*, non-periodic Gaussian data stream). It uses the same setup as Table 2. We'll clarify this in the caption of Table 1.
>
>
> > It is not specified whether a replay buffer is used to rehearse the previous knowledge (like in DER) or not (like in LwF)
>
> $\to$ We use the replay buffer. For managing the buffer, we used Greedy Balancing Sampler (Prabhu et al., 2020). For training the model with the memory, we use memory-only training (Koh et al., 2021) where we train only using randomly selected samples from memory. We clarify this detail in Sec. 6 in the revision.
>
>
> > Visualize how the loss balancing parameter change over time.
>
> $\to$ Thank you for the good suggestion! We have added the plot as Fig. 6 in Sec. A.8 of the revision. The balancing parameter generally starts from 1 and decreases as the training progresses.

---

> ### Author Response · Authors · 2022-11-12
> **Answers to the questions of Reviewer 46rh (2/4)**
>
> > The proposed metrics seem to be overly-complicated. If I understood Section 4 correctly, at the end those information-theory-based metrics go down to "average accuracy loss over some set of classes" or "average accuracy gain over some set of classes". Then, why not measure these values directly? It would be both more intuitive and have better scaling (one always know the lower/upper bound for them).
>
> $\to$ The proposed metrics seem similar to the accuracy-based metrics but they are different from the accuracy-based measures in several aspects as follows, and for these differences, we respectfully argue that the proposed metrics are still necessary:
>
> First, the "average accuracy loss/gain over some set of classes" captures the joint effect of the gained knowledge and the lost knowledge over time, so it cannot measure the contribution of knowledge loss and knowledge gain separately. For example, if accuracy for the class did not change, *i.e.*, accuracy gain and loss are both zero, it does not necessarily mean that model has learned nothing and forgotten nothing. The model may have learned some new knowledge about the class, but also forgotten some previously learned knowledge about the class so that the accuracy for the class did not change. Instead, we measure the gained/lost information separately using information theory, by the difference between total information in both (past and present) models and information in one model. For example, if the total information in both past and present models is 6 bits and information in the present model alone is 4 bits, we know that 2 bits of knowledge were lost. If the information in the past model is 3 bits, we know that 3 bits of knowledge were gained. Note that total information is not equal to the sum of two individual pieces of information, as there can be redundant information in two models.
>
> Second, accuracy is affected by the bias in the model (*i.e.*, model's tendency to predict towards some classes without learning meaningful information -- please refer to our answer to your previous question), so it does not accurately represent the amount of knowledge in a model. For example, if the model at $t_1$ predicts all samples as class 0 and the model at $t_2$ predicts all samples as class 1, class 0 would have 100% accuracy loss and class 1 would have 100% accuracy gain, so the average gain and loss in accuracy will be both 50%. However, both models have zero knowledge about the data, *i.e.*, we do not get any information from knowing the model output. In contrast, entropy of both models is calculated as zero, so knowledge gain and knowledge loss are measured as zero, which makes more sense.
>
> For the scale, same as the accuracy-based measures, KLR and KGR are also bounded in [0, 1]. Since KLR and KGR are defined as the measured KL and KG divided by the maximum possible value of KL and KG at this point, their upper bounds are 1. In addition, they have lower bounds of 0 because the KL and the KG are computed by conditional mutual information, which is always non-negative. Note that in Tables 2 and 3, their range is [0, 100] because we show percentage values of KLR and KGR (*i.e.*, multiplied by 100), alongside the accuracies.

---

> ### Author Response · Authors · 2022-11-12
> **Answers to the questions of Reviewer 46rh (1/4)**
>
> > It's true that there is a lot of real-world data is periodic but how improving on the proposed periodic benchmarks can improve the performance of some real-world system which operates on periodic data. Provide me an example of some real-world system which operates on periodic data and suffers from forgetting.
>
> $\to$ A real-world application is an automatic image-based shopping item category registration system for an e-commerce platform company whose search data are variously periodic over the seasons. Although the company is using one of the state of the art continual learning methods (DualNet, Pham et al., 2021) to actively update the model, they are experiencing quite severe forgetting, compared to the ‘joint training’ (training the model with the full data that is obtained by augmenting the old data and new data). So, while they update the model actively by the DualNet for some time (*e.g.*, six months -- having two seasons in it), they also use additional computing resources for joint training in the background, and regularly deploy the jointly trained model for better performance (*e.g.*, once every six months). We expect to promote research in this direction by the proposed benchmark to develop a continual learning system that is practically more useful for real-world applications.
>
> > In terms of complexity/performance trade-off, just using EMA without learning the balancing coefficient looks better to me.
>
> $\to$ Thank you for the interesting suggestion! We are also curious about the results and conduct experiments to measure the accuracy as a function of GPU hours (computational complexity), and summarize the results in Figure. 5 in Sec. A.6 in the revision. As shown in the figure, we observe the trade-off between the GPU hours and accuracy ($A_{AUC}$ (left) and $A_{last}$ (right)). We cordially argue that the proposed SDP has a better trade-off than the EMA without learning the balancing coefficient (‘EMA-only’ in Fig. 5); SDP improves $A_{AUC}$ and $A_{last}$ noticeably ($A_\text{last}$:+9.74% $A_\text{AUC}$:+2.76%) over EMA-only with little increase in training time (+0.14 hr).
>
> > The proposed metrics are not well-motivated. It is not clear why $A_\text{AUC}$ is bad, what important properties do KLR and KGR measure? Why should we measure KLR and KGR, they seem to always correlate with AAUC anyway
>
> $\to$ While $A_\text{AUC}$ measures the overall performance of the method, it mixes up the effect of stability and plasticity of a model (but cordially note that we have not claimed that the $A_\text{AUC}$ is bad). To separately investigate the stability and plasticity of the model, we propose the KLR and KGR. Although there are existing metrics for measuring forgetting and intransigence (forgetting (F) and intransigence (I) proposed by Chaudhry et al., 2018a) for this purpose, we propose the KLR and KGR as they have the following weaknesses:
>
> 1. Forgetting (F) and Intransigence (I) are defined based on the task. So, they cannot be computed under the boundary-free setup.
>
> 2. The F and the I are measured by the increase or decrease of the accuracy of each class. Since the accuracy of each class measures the combined effect of knowledge gain and knowledge loss, if the loss and gain of knowledge occur simultaneously in the class, they cannot be captured by F and I.
>
> 3. Since the F and the I use accuracy to measure loss and gain of knowledge, *bias of the model* might be also measured as knowledge. Here, we mean the model’s bias as follows; for example, if the model predicts all samples as class 0, the accuracy for class 0 becomes 1 and is treated as having perfect knowledge of class 0 by the existing forgetting metric. However, the model does not provide any meaningful information about class 0. In contrast, KLR and KGR measure the knowledge gain and loss by 0 since the model’s entropy is 0 (they are measured by the mutual information).
>
> **Correlation of KLR and KGR to $A_{AUC}$**: Yes, KLR and KGR are always inevitably correlated with $A_\text{AUC}$ since both the stability and plasticity affect the overall performance of a method. But we respectfully argue that the KLR and KGL are useful as evaluation metrics as they factorize the effect of stability and plasticity of a model for accuracy. For example, in Tables 6 and 7, KLR is much lower in the periodic data stream than the non-periodic data stream on CIFAR-10. While $A_{AUC}$ increases, there is not much difference in KGR, indicating that the improvement in KLR (*i.e.*, less forgetting) is accountable for the higher overall performance ($A_{AUC}$). However, this gap between periodic and non-periodic decreases for larger datasets (*e.g.*, in TinyImageNet and ImageNet there are almost no differences). This implies that while periodicity of the data may alleviate forgetting on small-scale data, this effect decreases as the number of classes and number of samples increase, and periodicity of data alone may not alleviate forgetting in large-scale systems.

---

### Official Review · Reviewer_VkR2 · 2022-11-03

**Confidence:** 3
**Correctness:** 3
**Technical Novelty And Significance:** 3
**Empirical Novelty And Significance:** 2
**Recommendation:** 8

**Clarity, Quality, Novelty And Reproducibility:**

The novelty of this paper is ok

Enough details for reproducibikity

**Strength And Weaknesses:**

Advantages:



This paper has a very strong mathematical background.

Novel Setup, metrics and losses have been proposed for online boundary-free continual learning problem.

The method has been compared to state-of-the-art papers and achieved better results.


Disadvantages:



It is preferable if a more detailed explanation is provided about the knowledge distillation term of loss function formula 9 on the page and its direct relation with formula 7 on page 5.

H(Y GT) on page 4 has not been defined and it is only mentioned it is [the amount of information in GT.

A_AUC and A_last in table 1 on page 6 are defined afterward on page 7

**Summary Of The Paper:**

This paper addresses the continual learning problem in a more realistic situation by considering the fact that the real-world data is online and has no explicit boundaries and its distribution shifts over time. A novel setup for online task boundary-free has been proposed which modeled the arrival time of data by periodic Gaussian distribution. Then two new evaluation metrics have been proposed based on information theory and the concept of mutual information. These metrics do not depend on old and new tasks and satisfy the online condition. To train their model, they suggest a two-term loss consisting of Cross-Entropy loss for current data samples and a Knowledge distillation Loss for past samples which combines the data by a novel method using hypo-exponential distribution called scheduled data prior (SDP) which is a weighted summation of two exponential moving average functions. The method is claimed to work on both online and task-specific setups and improve the metrics by a large margin.

**Summary Of The Review:**

The paper is well written and also have a good novelty

---

> ### Author Response · Authors · 2022-11-12
> **Answers to the questions of Reviewer VkR2**
>
> > More detailed explanation about the knowledge distillation term of loss function formula 9 on the page
>
> $\to$ The knowledge distillation term $\mathcal{L}_\text{KD}(x) = \|f(x) - f_\text{SDP}(x)\|^2_2$  is an L2 loss between the online model $f(x)$ and the SDP model $f_\text{SDP}(x)$.
>
> > > and its direct relation with formula 7 on page 5
>
> $\to$ The SDP model $f_\text{SDP}(x)$ in knowledge distillation term is a neural network with parameters $\theta_{\text{SDP}(\alpha, \beta)}$ obtained by the Eq. 7 on page 5.
>
> We clarify these in Sec. 5 of the revision. Thank you for the suggestion.
>
>
> > $H(Y_{GT})$ on page 4 has not been defined and it is only mentioned it is [the amount of information in GT].
>
> $\to$ $H(Y)$ is the entropy of $Y$, defined as $H(Y)=-\sum_{y\in \mathcal{Y}}P(y) \log P(y) $. We add the definition in Sec. 4 of the revision. Thank you.
>
>
> > A_AUC and A_last in table 1 on page 6 are defined afterward on page 7
>
> $\to$ Thank you! We move the definition to Sec. 4 in the revision.

---

> > ### Comment · Reviewer_VkR2 · 2022-11-22
> > **upgrade my rate**
> >
> > Thank you for answering my concerns. The paper has enough value to publish in ICLR

---

> ### Author Response · Authors · 2022-11-18
> **Discussion reminder**
>
> We sincerely thank you for your effort in reviewing our submission. We gently remind the reviewer that we tried our best to address your concerns via our replies and revision of the manuscript. As the discussion period is nearing the end, we would be delighted to hear more from you if there are any further concerns.

---

### Author Response · Authors · 2022-11-12
**General response**

We thank the reviewers for their helpful feedback and encouraging comments including the novel, more realistic (practical), important or challenging setup (**VkR2, 46rh, 1ivp, Y7gj**), novel metrics (**VkR2, 46rh, 1ivp, Y7gj**), new losses (**VkR2, 46rh**), well motivated data-driven balancing (**Y7gj**), simplicity of approach (**46rh**), strong mathematical background (**VkR2**), empirical results being comprehensive (**46rh**), significant improvement (**1ivp**) and outperforming state of the arts (**VkR2**), detailed ablation study (**1ivp**), and clear presentation (**VkR2, 46rh**).

We have uploaded the first revision of the manuscript (changes are highlighted by blue color). Note that hyperparameter search results have been moved from Sec. 6.4 to Sec. A.9 in the appendix to accommodate the additional information added in the revision.

---

### Author Response · Authors · 2022-11-18
**Second revision**

We have uploaded the second revision of the manuscript. The revision includes additional experiments with new setups, to address the reviewers’ concerns and suggestions (**1ivp**, **46rh**). SDP outperforms all other methods by significant margins on these setups.

Summary of the changes
- Add ablation study on the number of periods (**46rh**) in the Periodic Gaussian data stream in A.13.
- Add experiments on more complex Gaussian Mixture distributions (**1ivp**, **46rh**) in A.14.
- Add accuracy trends graph for the tested CL methods in A.15.
- Add the asymptotics analysis on memory cost (**46rh**) in A.7.

---

### Author Response · Authors · 2022-11-18
**Third revision**

We have uploaded the third revision of the manuscript. In the revision, we add additional experiments on CLEAR-10 dataset in A.16, to further address **46rh**’s concern on our benchmark. SDP again outperforms all other methods by significant margins in the new experiments.

---

### Decision · Program_Chairs · 2023-01-20

**Decision:**

Accept: poster

**Justification For Why Not Higher Score:**

The paper proposes a clear idea to solve a non-trivial problem within continual learning. I believe it warrants acceptance and publication. However, the setting is still quite niche and only applies to specific sub-group of continual learning community. It is not general enough to warrant an oral or spotlight presentation.

**Justification For Why Not Lower Score:**

The paper has clearly a merit to be presented which has been agreed with the reviewers as well. Overall, it is well-written about a technically sound idea with a solid empirical study and promising results. I believe it should be shared with community and I do not see any possible major improvement if it was asked to be resubmitted.

**Metareview: Summary, Strengths And Weaknesses:**

The paper is proposing a method to tackle online continual learning. The setup is truly online in a sense that it uses streaming data and processes them one by one. This is a much more realistic setting which have recently been studied. I appreciate that this more realistic setup is getting more popular. The proposed approach is based on learning two models of current information and past information with a later distillation process. The model representing historical data is learned using EMA with a novel weighting function designed for periodic data and peaks somewhere in history instead of current or oldest point. It is more of a repetition within a fixed window framework which has support from other fields like education. The paper has been reviewed by 5 expert reviewers among 4 agrees on the merit and acceptance. The remaining reviewer did not raise any significant issue as the majority of the review is about lack of clarity around EMA which I believe authors should improve in the camera ready. Specifically, a detailed introductory discussion on EMA in the main text would be beneficial.

**Note From Pc:**

if the above contains the word "oral" or "spotlight" please see: "oral" presentation means -> notable-top-5% and "spotlight" means -> notable-top-25%. As stated in our emails, we are disassociating presentation type from AC recommendations